# A flavin-monooxygenase catalyzing oxepinone formation and the complete biosynthesis of vibralactone

Ke-Na Feng [1,6], Yue Zhang [1,5,6], Mingfang Zhang[2,5,6], Yan-Long Yang[1,4], Ji-Kai Liu[3], Lifeng Pan [2]✉ & Ying Zeng [1]✉

Oxepinone rings represent one of structurally unusual motifs of natural products and the biosynthesis of oxepinones is not fully understood. 1,5-*Seco*-vibralactone (**3**) features an oxepinone motif and is a stable metabolite isolated from mycelial cultures of the mushroom *Boreostereum vibrans*. Cyclization of **3** forms vibralactone (**1**) whose β-lactone-fused bicyclic core originates from 4-hydroxybenzoate, yet it remains elusive how 4-hydroxybenzoate is converted to **3** especially for the oxepinone ring construction in the biosynthesis of **1**. In this work, using activity-guided fractionation together with proteomic analyses, we identify an NADPH/FAD-dependent monooxygenase VibO as the key enzyme performing a crucial ring-expansive oxygenation on the phenol ring to generate the oxepin-2-one structure of **3**. The crystal structure of VibO reveals that it forms a dimeric phenol hydroxylase-like architecture featured with a unique substrate-binding pocket adjacent to the bound FAD. Computational modeling and solution studies provide insight into the likely VibO active site geometry, and suggest possible involvement of a flavin-C4a-OO(H) intermediate.

Vibralactone (**1**, Fig. 1) is a rare 4/5 fused bicyclic β-lactone decorated with a dimethylallyl group and first isolated from mycelial cultures of *Boreostereum vibrans* (syn. *Stereum vibrans*, a basidiomycete fungus)[1] and other *Stereum* mushrooms afterwards[2–5], along with 1,5-*seco*-vibralactone (**3**) featuring a unique oxepin-2(3*H*)-one ring[6]. Vibralactone was reported to inhibit the pancreatic lipase significantly relevant to fat absorption with an IC$_{50}$ value of 0.4 μg·mL$^{-1}$, comparable to that of the anti-obesity drug orlistat (IC$_{50}$ = 0.18 μg·mL$^{-1}$)[7]. Chemical biology studies revealed that vibralactone targets both ClpP1 and ClpP2 of caseinolytic peptidases that are highly conserved in the pathogenic bacterium *Listeria monocytogenes* and vital for its virulence, unlike most β-lactone compounds that bind solely the ClpP2

isoform[8]. Particularly, vibralactone can block the activities of acyl-protein thioesterases involved in Ras signaling[9]. Inspired by its unusual scaffold and bioactivity, synthetic chemists developed multiple routes for the total synthesis of vibralactones[10–13].

Vibralactone has a unique chemical structure and potent biological activity, which make vibralactone a worthy target for biosynthetic investigation. Based on initial incorporation studies with [U-$^{13}$C]4-hydroxybenzoate and analog intermediate feedings to the *B. vibrans* growing culture, we established that (i) the bicyclic lactone core of vibralactone (**1**) is derived from an aryl ring, and 4-hydroxybenzoate (**2**) serves as the direct ring precursor of vibralactone[14]; (ii) 3-prenyl-4-hydroxybenzylalcohol (**6**) is an on-pathway biosynthetic intermediate

[1]State Key Laboratory of Phytochemistry and Plant Resources in West China and Yunnan Key Laboratory of Natural Medicinal Chemistry, Kunming Institute of Botany, Chinese Academy of Sciences, Kunming 650201, China. [2]State Key Laboratory of Chemical Biology, Shanghai Institute of Organic Chemistry, University of Chinese Academy of Sciences, Chinese Academy of Sciences, Shanghai 200032, China. [3]School of Pharmaceutical Sciences, South-Central Minzu University, Wuhan 430074, China. [4]College of Chemistry and Chemical Engineering, Lanzhou University, Lanzhou 730000, China. [5]University of Chinese Academy of Sciences, 100049 Beijing, China. [6]These authors contributed equally: Ke-Na Feng, Yue Zhang, Mingfang Zhang. ✉e-mail: panlf@sioc.ac.cn; biochem@mail.kib.ac.cn

**Fig. 1 | Biosynthetic pathway of vibralactone (1).** This study demonstrates that 4-hydroxybenzoate (**2**) is converted by the UbiA prenyltransferases (VibP1/VibP2) into **4**, which is reduced stepwise to the corresponding aldehyde (**5**) by catalysis of the carboxylic acid reductase *Bv*CAR and the following benzylic alcohol (**6**) by aldehyde reductases (*Bv*ARs). Subsequently, **6** undergoes ring-expansive oxygenation catalyzed by an NADPH/FAD-dependent monooxygenase VibO to generate the oxepinone unit in **3**. Cyclization of **3** to vibralactone (**1**) by the action of VibC was found before[16].

for forming the 1,5-*seco*-vibralactone (**3**) and vibralactone (**1**)[15]. Furthermore, we revealed the enzymatic and structural bases for the last step of cyclization of **3** to **1**, which is catalyzed by a cyclase VibC[16]. Yet, it remains unknown which enzymes are responsible for producing **3** from the precursor 4-hydroxybenzoate (**2**), especially the oxepinone ring construction to afford **3** from **6** as well as the intermediary for the conversion of **2** to **6** in the biosynthesis of vibralactone. We speculated that sequential steps might proceed via a biosynthetic route as **2**→**4**→**5**→**6** (Fig. 1) and **6** may undergo oxygenation of its phenol ring to give the oxepinone motif of **3**. Notably, oxepinone rings are unusual motifs and only a few oxepinone-bearing compounds have been identified in nature, which display remarkable biological activities[3,17–20]. Enzymes responsible for constructing some of the oxepinone rings are multifunctional to catalyze consecutive oxygenations and decarboxylation, exemplified by the flavoenzyme GilOII for the gilvocarcin pathway[21] and the cytochrome P450 OrdA[22] for the aflatoxin biosynthesis. As the vibralactone pathway features an oxepin-2-one ring construction leading to **3** as a stable metabolite in the *B. vibrans* mycelial cultures, we attempted to biochemically and structurally elucidate the enzymes governing the oxepinone-formation of vibralactone.

In this work, we uncover that four enzymes are required for the generation of **3** from 4-hydroxybenzoate (**2**) by the intermediacy of 3-prenyl-4-hydroxybenzoate (**4**) via aldehyde **5** to alcohol **6**, and demonstrate, for the first time, that the oxepinone-forming enzyme VibO serves as an NADPH/FAD-dependent monooxygenase to catalyze ring-expansive oxygenation on the phenol unit of **6** to generate the unique oxepin-2(3*H*)-one motif of **3** as its direct product. Interestingly, hydroxylation of **6** to a side product **7** harboring a catechol moiety occurs in the VibO-catalyzed reactions supplied with sufficient NADPH. Despite the high structural similarity between VibO and the phenol hydroxylase, our biochemical and structural results indicate that VibO catalyzes the formation of oxepinone through Baeyer-Villiger oxidation. Furthermore, we also characterize VibP1 and VibP2 that perform as membrane-bound prenyltransferases, together with reductases *Bv*CAR and *Bv*AR for the conversion of **2** to **6**. Finally, the complete five-enzyme vibralactone pathway has been functionally reconstituted both in vitro and in *Escherichia coli*.

## Results and discussion
### Identification of the oxepinone-forming enzyme through the activity-guided fractionation
As vibralactone biosynthetic genes unlikely occur as a gene cluster[16], we set out to conduct mycelia enzyme fractionation instead of the direct gene isolation. According to the activity-guided fractionation combined with proteomic analyses, which was successfully used to discover the vibralactone cyclase VibC[16], we start with the *B. vibrans* mycelia that significantly accumulate **3** in the culture broth. The fractionation was guided by a specific enzyme assay using **6a** (analog of **6**, Supplementary Fig. 1) that harbors an allyl rather than dimethylallyl group to recognize the assay-formed product from the native **3** in fungal mycelia, due to a small amount of endogenous **3** that may still exist in fractions even after ion-exchange chromatography. Sequential

chromatography led to seven fractions enriched in the enzyme activity, which were further individually subjected to liquid chromatography-tandem mass spectrometry (LC-MS/MS) for proteomic identification. This yielded 69 proteins common to all the seven active fractions, among which 19 candidates were selected according to the apparent molecular weight (~60 kDa) of the active protein on the size-exclusion chromatography (Supplementary Figs. 2 and 3, Source Data). Fourteen candidates were successfully cloned from complementary DNA (cDNA) of *B. vibrans*, expressed in *Escherichia coli* BL21(DE3), and assayed for enzyme activity by incubating the relevant cleared cell lysate with **6a** as substrate. The reaction was analyzed by LC-MS for the extracted ion chromatogram (EIC) of mass-to-charge ratio (*m/z*). Among the 14 candidates screened, two putative flavin-binding monooxygenases with 99% sequence identity, were found to produce a compound with a $[M + H]^+$ of 181 *m/z*, corresponding to the authentic **3a** (Supplementary Fig. 3, Supplementary Table 2). Furthermore, biochemical assays using **6** as substrate revealed a product consistent with the authentic **3** with a $[M + Na]^+$ of 231 *m/z*, in comparison to control assays using the lysate of *E. coli* cells transformed with an empty pET28a(+) vector (Supplementary Fig. 4). Likewise, their homologs (95% identity, accessions. XP_007301038.1 and XP_007301015.1) amplified from cDNA of the mushroom *Stereum hirsutum*, which can generate vibralactone as a minor metabolite[5], and expressed in *E. coli* can also produce **3** from **6** as substrate (Supplementary Fig. 4). The enzyme, which was purified to homogeneity as an N-terminal His$_6$-tag protein, was shown to yield **3** as a significant product when incubating with **6** in the presence of the exogenous NADPH (nicotinamide adenine dinucleotide phosphate, reduced), while no detectable **3** can be observed in the reaction with NADH or in control assays with boiled enzyme (Fig. 2a, Supplementary Figs. 3 and 5). To obtain enough product for structural elucidation, we conducted product extraction and purification from a whole-cell biotransformation by feeding 90 mg of **6** to the *E. coli* strain expressing the enzyme, which led to the isolation of approximately 12 mg of the product. Its identity was validated to be the oxepin-2(3*H*)-one-containing (1 *S*)-1,5-*seco*-vibralactone (**3**) by evidence of $^1$H NMR spectrum and optical rotation, matching the authentic **3** isolated from the *B. vibrans* mycelial cultures. Formation of **3** by the purified enzyme rose onefold from 10 to 60 min, as determined by quantification analysis of time-course reactions (Supplementary Figs. 6–8). Considering that **3** and its oxepin-2(7*H*)-one isomer **3′** can tautomerize from each other[16], LC-MS detection was performed immediately after incubation without product extraction and drying overnights. Results confirmed **3** as the direct product of the enzymatic catalysis (Fig. 2b, Supplementary Fig. 9). Thus, the oxepinone-forming enzyme responsible for conversion of **6** into **3** was identified and named as VibO, in line with VibC in the same pathway for the biosynthesis of vibralactone (**1**).

Alongside production of **3** from **6** by the catalysis of VibO, a very small peak (4.36 min, Fig. 2a) was found on the same EIC trace for +*m/z* 231 but corresponding neither to **3** nor **3′**. The peak was more noticeable on EIC for −*m/z* 207 where no signals for **3** and **3′** were observed. In the reaction with 1 mM NADPH, slightly more production

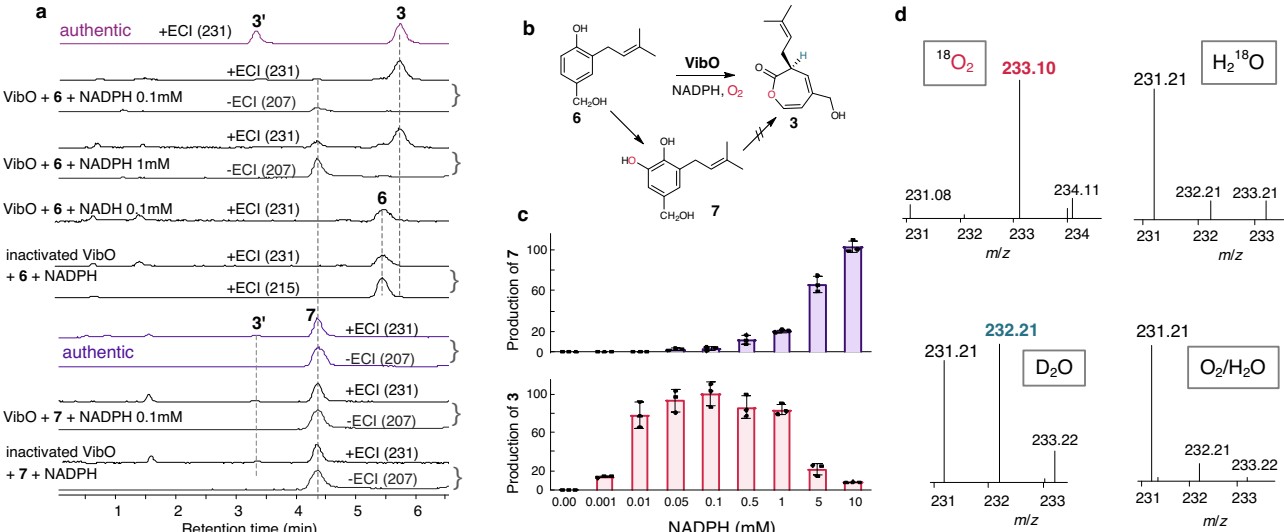

**Fig. 2 | Biochemical characterization of the flavin-dependent monooxygenase VibO expressed in *E. coli* as a soluble protein. a** LC-MS analyses of in vitro activities of VibO, using the reaction of heat-inactivated enzyme as control. The extracted ion traces of $m/z$ 231 are corresponding to $[M + Na]^+$ of **3, 3′, 7** and $[M + K]^+$ of **6**, respectively; $m/z$ 215 is for **6** $[M + Na]^+$; $m/z$ 207 is for **7** $[M–H]^-$. **3′** is an oxepin-2(7*H*)-one isomer of **3** (see Fig.6a). A trace of **3′** remains as impurity in isolation of compound **7**. **b** Reaction schemes of VibO. **c** Dependence of **3** and **7** production on NADPH (0 to 10 mM). The yield was estimated from peak area ratios of the product to the internal standard. The columns represent accumulation of products (the highest mean value was set 100%); bars indicate ± SD (standard deviation) of three replicates. Source data are provided as a Source Data file. **d** Mass spectra of VibO-formed **3** in $^{18}O_2$/$H_2O$, $O_2$/$H_2^{18}O$, $O_2$/$D_2O$ (deuterated), and the normal $O_2$/$H_2O$ buffer systems. The original data are provided in Supplementary Figs. 5–11, 15–18.

of this side-product was observed (Fig. 2a, Supplementary Fig. 5). To determine its chemical structure, a scale-up in vitro reaction was conducted since isolation of the side-product was unsuccessful from the *E. coli* whole-cell transformation due to its low production in this in vivo system (see below). Consequently, the side-product was identified by $^1H$ NMR spectrum as the compound **7** which has a catechol nucleus; its data were consistent with those of earlier publication[23] (Fig. 2b, Supplementary Fig. 10). To examine the dependence of NADPH, reactions of VibO and **6** with NADPH ranging from 0.001 to 10 mM were detected by EIC +$m/z$ 231 for both **3** and **7**. Production of **3** was observed at 0.001 mM NADPH and increased to the highest yield at 0.1 mM NADPH, then drastically decreased by ~12-fold at 10 mM NADPH. In contrast, NADPH up to 10 mM resulted in the highest production of **7** with a ~40-fold increase compared to its initial formation at 0.05 mM NADPH (Fig. 2c, Supplementary Fig. 11). The increase in **7** can hardly be attributed to the decrease in **3**, as proved by incubations of **3** with NADPH in which no trace of **7** was found (Supplementary Fig. 12). Moreover, when **7** was used as a substrate and incubated with VibO, no detectable **3** was observed (Fig. 2a, b; Supplementary Fig. 5). The compound **7** was barely detected in the native *B. vibrans* fungal cultures, and just a small amount of **7** was observed in the *vibO*/pET28a/*E. coli* strain fed with **6** for 24 h. Conversely, accumulation of **3** was obvious in both of the in vivo systems (Supplementary Figs. 13 and 14).

To determine the origin of the inserted oxygen atom and hydrogen exchange in **3**, isotope-labeling experiments using $^{18}O_2$, $H_2^{18}O$, or $D_2O$ (deuterated $H_2O$) in the VibO reaction were performed with comparison to the normal $O_2$/$H_2O$ system. Isotopologue ions ($m/z$ 233) for **3** shifted by 2 $m/z$ were significantly observed when the VibO reaction was conducted under $^{18}O_2$ atmospheric conditions, indicating that one $^{18}O$ atom from molecular oxygen was incorporated into **3** (Fig. 2b, d). The $^{13}C$ NMR spectrum of labeled **3** further confirmed the presence and position of $^{18}O$ atom (Supplementary Figs. 15 and 16). This $^{18}O_2$ condition also gave $^{18}O$-labeled **7** at $m/z$ 233 (Supplementary Figs. 17 and 18). Meanwhile, the occurrence of $m/z$ 232 for **3** in $D_2O$ system suggested one hydrogen exchange with moderate deuterium incorporation during the VibO catalysis, which was supported by

evidence of $^1H$ NMR spectrum (Supplementary Figs. 6 and 7). However, **7** was barely labeled in $D_2O$ system (Supplementary Figs. 17 and 18). When the assay was conducted in $H_2^{18}O$ (90% enriched), no $^{18}O$ incorporation into **3** was detected (Fig. 2d). The VibO solution displays a yellow color, indicating the presence of flavin. Further LC-MS analysis of the denatured enzyme confirmed the bound flavin of VibO as flavin adenine dinucleotide (FAD), which is consistent with the finding that VibO has almost the same level of activities with or without the exogenous FAD and requires no additional FAD for in vitro assays (Supplementary Fig. 19). The freshly purified VibO has an apparent $K_M$ value at $0.407 \pm 0.083$ mM toward **6** and a turnover number ($k_{cat}$) at $0.111 \pm 0.010$ min$^{-1}$ (Supplementary Fig. 8). VibO is dependent exclusively on NADPH as no detectable product was observed with NADH as the reductant (Fig. 2a, Supplementary Fig. 5). Concomitant release of $H_2O_2$ was hardly observed in VibO reactions (Supplementary Fig. 20).

VibO has the conserved sequence motifs GXGXXG and GD that are involved in FAD binding, and contains the DG fingerprint that is important for binding to NAD(P)H (Supplementary Fig. 21). Phylogenetic analysis revealed that VibO belongs to group A flavin-dependent monooxygenases and is clustered with type O Baeyer-Villiger monooxygenases and aromatic ring hydroxylases (Supplementary Fig. 22). In this group, GilOII[21], MtmOIV[24], FlsO1[25], XanO4[26] and TerC[27] are multifunctional flavoenzymes to perform consecutive oxygenation (hydroxylation/epoxidation/Baeyer-Villiger oxidation) and decarboxylation upon binding specific substrates (Supplementary Fig. 23). VibO, on the other hand, acts as an NADPH/FAD-dependent monooxygenase catalyzing oxygenation on the phenol ring of **6** to form the oxepinone **3** in the biosynthetic pathway to vibralactone (**1**).

## Crystal structure of VibO

To gain molecular insights into the enzymatic mechanism of VibO, the crystal structure of VibO in complex with the cofactor FAD was determined to 2.43 Å resolution (PDB ID: 7YJ0 [https://doi.org/10.2210/pdb7YJ0/pdb]), (Supplementary Table 1). In the crystal structure, each asymmetric unit contains four VibO molecules (Supplementary Fig. 24a), which form two symmetric dimers (Fig. 3b), in line with our analytical ultracentrifugation-based assay (Supplementary

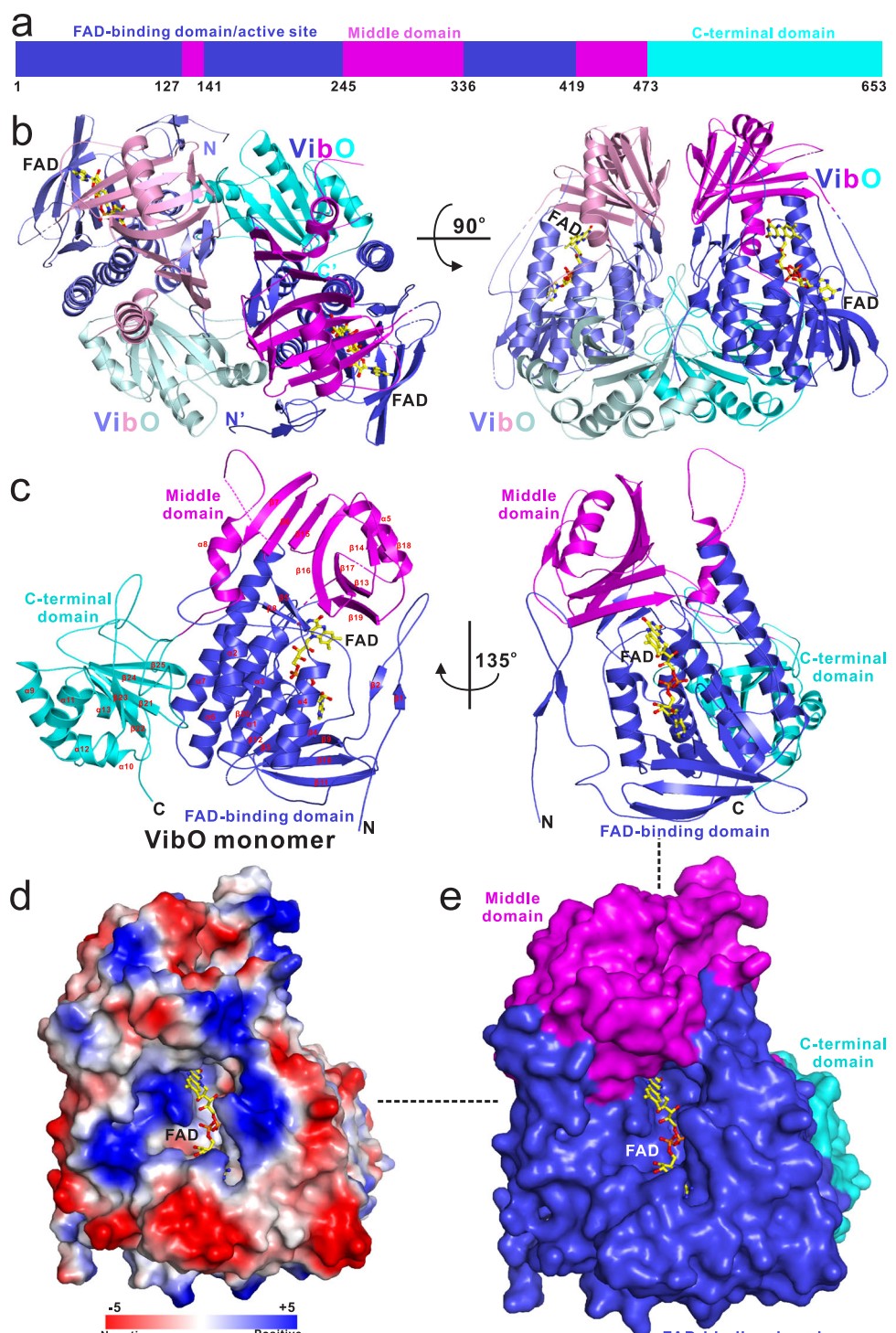

**Fig. 3 | The overall structure of VibO. a** A schematic diagram showing the domain organization of VibO. **b** The ribbon-stick representation showing the overall structure of the dimeric VibO. The FAD-binding domain, the middle domain, and the C-terminal domain of VibO are shown in blue/slate, magentas/pink, and cyan/pale cyan, respectively. The two bound cofactor FAD molecules are shown in the stick mode. **c** The ribbon-stick representation showing the overall architecture of the monomeric VibO with the cofactor FAD. The ribbon diagram representation uses the same color scheme as that in **b**. **d** The combined surface charge potential representation (contoured at ±5 kT/eV; blue/red) and the stick-ball model showing the solvent-exposed and highly positive-charged FAD-binding site of VibO. **e** The combined surface representation and the stick model showing the location of the FAD-binding site in the VibO structure.

Fig. 24b). In the dimeric VibO structure, each VibO monomer contains 25 β-strands (β1-β25) and 13 α-helices (α1-α13) (Fig. 3b, c), and is composed of three distinct sub-domains: the FAD-binding domain (residues 1–127, 141–245 and 336–419), the middle domain (residues 128–140, 246–335 and 420–472), and the C-terminal domain (residues 473–653) (Fig. 3a, c). In particular, the FAD-binding domain is

mainly assembled by an N-terminal extension (residues 1–56) and a central three-layer (ββα) sandwich that is formed by a β-sandwich packing with an overlaying helical section (Fig. 3c). The middle domain is not continuous in amino acid sequence (Fig. 3a), and folds into an architecture featured with a seven-stranded β-sheet flanked by two α-helices (α5 and α8) together with a short antiparallel β-sheet

(β14 and β18) (Fig. 3c). The C-terminal domain of VibO adopts a peroxiredoxin-like α/β-fold, which directly packs against the FAD-binding domain and is far away from the middle domain (Fig. 3c). The overall architecture of VibO is similar to that of the flavin-dependent phenol hydroxylase PHHY (PDB ID: 1PN0)[28,29] (Supplementary Fig. 25a), which catalyzes the *ortho*-hydroxylation of phenol to catechol and belongs to the group A flavoprotein monooxygenase family[30], as revealed by a structural similarity search using the program Dali[31]. Furthermore, the overall structure of VibO is similar to the type-O Baeyer-Villiger monooxygenase MtmOIV (PDB ID: 4K5S) (Supplementary Fig. 25b), but distinct from that of type-I Baeyer-Villiger monooxygenases, such as the fungal Baeyer-Villiger oxygenase BVMO_AFL838 (PDB ID: 5J7X), which lacks the C-terminal Rossmann fold sub-domain (Supplementary Fig. 25c). However, the dimer assembly mode of VibO is distinct from that of PHHY and MtmOIV (Supplementary Fig. 25d, e). In particular, the dimerization of VibO is mediated by three different types of sub-domain/sub-domain interactions between the two monomeric VibO molecules (the middle domain/middle domain interaction, the FAD-binding domain/FAD-binding domain interaction and the FAD-binding domain/C-terminal domain interaction) (Fig. 3b, Supplementary Fig. 26), covering a total of ~3436 Å$^2$ surface area. Further detailed structure analyses revealed that the dimerization interface of VibO is mainly mediated by extensive polar (charge-charge and hydrogen bonding) and hydrophobic interactions (Supplementary Fig. 26).

Across the middle and FAD-binding domains of VibO, the cofactor FAD is buried in a solvent-exposed and highly positive-charged pocket (Fig. 3d, e). Specifically, the ADP and ribityl groups of the bound FAD are embedded within the FAD-binding domain, whereas its isoalloxazine ring is located at the interface between the middle domain and the FAD-binding domain (Fig. 3e). Notably, the co-factor FAD in the crystal structure of the VibO/FAD complex adopts an "out"-conformation (Supplementary Fig. 25f), suggesting that the bound FAD is located outside of the active site of VibO. Further careful analyses of the VibO structure uncovered that hydrogen bonds and electrostatic interactions are primarily responsible for the specific binding of VibO with FAD, involving the main chains of G66, V68, S89, D356, A363 and M369 as well as the side chains of R88, R97, Q160, R287 and D356 of VibO (Supplementary Fig. 27).

Next, we sought to obtain a complex structure of VibO with the substrate or product. But, unfortunately, despite numerous attempts using co-crystallization or soaking method, none of the crystals analyzed showed reasonable electron density for the bound substrate or product, likely due to the poor solubility of the substrate in the buffer solution as well as the weak binding between VibO and the product. Careful structural analyses showed that there is a highly hydrophobic cavity adjacent to the isoalloxazine ring of the bound FAD in the VibO/FAD complex, and importantly, this pocket corresponds to the substrate-binding site of the phenol hydroxylase PHHY (Supplementary Fig. 28a). Using the Autodock Vina program[32], we conducted a structural modeling analysis of VibO with substrate **6**. A search for the lowest binding energy and the best complementary shape led to a reasonable conformer of substrate **6** (Fig. 4a), which accommodates the cavity well and forms favorable hydrophobic and polar interactions with the surrounding VibO residues, including D99, L155, A157, F265, I279, Y289, and A366 (Fig. 4a, b). Importantly, our site-directed mutagenesis-based analyses showed that point mutations of key binding residues of VibO, such as F265Y, I279F, Y289A, A366L and A366Q, all largely decreased the enzymatic activity of VibO (Fig. 5a), further validating our structural modeling result. Notably, as a result of "out"-conformation of FAD, the distance between the carbon C6 of substrate **6** and carbon C4a of FAD is 8.4 Å (Fig. 4b), longer than the reports of other reported FAD-dependent enzymes, which typically range from 4.5 to 5.5 Å[33,34]. Further structural comparison analyses uncovered that although the sizes and compositions of these pockets

in VibO and PHHY are somewhat different, the critical D54, I279, Y289 residues in PHHY for the hydroxylation activity are shared by VibO (Supplementary Fig. 28b). Strikingly, in contrast to PHHY, VibO contains a unique channel, which is located on the opposite side of the substrate-binding pocket and away from the FAD-binding site (Fig. 4a, Supplementary Fig. 29). Detailed structural characterizations elucidated that the substrate-binding pocket and the channel of VibO are formed between the middle domain and the FAD-binding domain, and are both assembled by hydrophobic and polar residues from the two domains (Supplementary Fig. 29b). Considering that the substrate-binding site of VibO is buried deep inside the structure (Fig. 4a), we speculated that this unique channel of VibO might be used for the substrate entrance or product leaving. Consistent with our hypothesis, the replacement of V459 that locates at the solvent-exposed site of the channel with a much larger Leu residue can nearly halve the enzymatic activity of VibO (Fig. 5a).

## Site-directed mutations and mechanism of VibO catalysis

Based on the structural analyses of the active-site of VibO (Fig. 4), the residues forming the substrate pocket including D99, F265, I279, Y289, and A366 were conducted for site-directed mutagenesis to reduce (D99N, Y289N) or even remove the hydrogen bond interaction (D99A, Y289A, and Y289F), or to narrow down the substrate-binding pocket (F265Y, I279F, A366L, and A366Q). Consequently, none of these variants can give the product **3** except for F265Y with a half decrease in accumulation of **3** compared to the wild-type enzyme (Fig. 5a). On the other side, only three variants of VibO including D99A, D99N, and F265Y were observed for production of **7** with catalytic activities of 16%, 10%, and 22%, respectively relative to the wild-type enzyme (Fig. 5a, Supplementary Figs. 30 and 31), which is consistent with a ~5-fold decrease in hydroxylation activity observed for the D54N variant of PHHY[35], as Asp54 in PHHY corresponds to the Asp99 residue in VibO (Supplementary Fig. 28b). The arginine variants of VibO completely lost the catalytic activity (Fig. 5a), which agrees with the proposed R residues (Fig. 4c, d) that can stabilize the negatively charged flavin-OO⁻ intermediate in the Baeyer-Villiger monooxygenase MtmOIV[24] or provide interactions for stabilizing the flavin ring in the hydroxylation of PHHY[28,35]. Corresponding to R97 and R287 of VibO, mutation of the FAD-binding residue R45 or R213 led to the inactivation of FlsO1[25] for which two reactive forms of flavin were proposed to perform Baeyer-Villiger oxidation (FAD-OO⁻) or hydroxylation (FAD-OOH) (Supplementary Figs. 21 and 23). Ultraviolet–visible spectral analysis of the effect of mutation revealed the inability of I279F and A366L to bind substrate **6** (Supplementary Fig. 32), emphasizing their key roles for construction of the substrate-binding pocket (Fig. 4b). The variant F265Y showed a lower capacity for binding the substrate and thus may contribute to its decreased catalytic activity (Fig. 5a, Supplementary Fig. 32). As the D99A variant had only marginal effect on the FAD binding and NADPH consumption (Supplementary Figs. 33 and 34), its inactivation to form the oxepinone **3** (Fig. 5a) may indicate a crucial role in the mechanism of VibO-catalyzed reaction where deuterium incorporation in **3** was obvious (Fig. 2d) and thus deprotonation of substrate **6** can be expected (Fig. 5d). Compared to the wild-type enzyme, the flavin absorption spectra of the substrate-free Y289A showed a slightly different peak mode and less FAD incorporation (Supplementary Figs. 32 and 33). Further examination on the substrate-induced difference spectra of Y289A revealed the maximum differential absorption at 383 nm rather than 490 nm as observed for others. These seem to hint about minor structural variations in the active site proximate to FAD (Supplementary Fig. 35), given Tyr289 residue as an essential structural element for the stabilization of FAD (Fig. 4d). Notably, almost identical levels of NADPH consumption over time by VibO were detected in the absence or presence of substrate **6** (Supplementary Fig. 34), supportive of the "out"-conformation of FAD observed for the VibO structure (Supplementary Fig. 25f). This finding

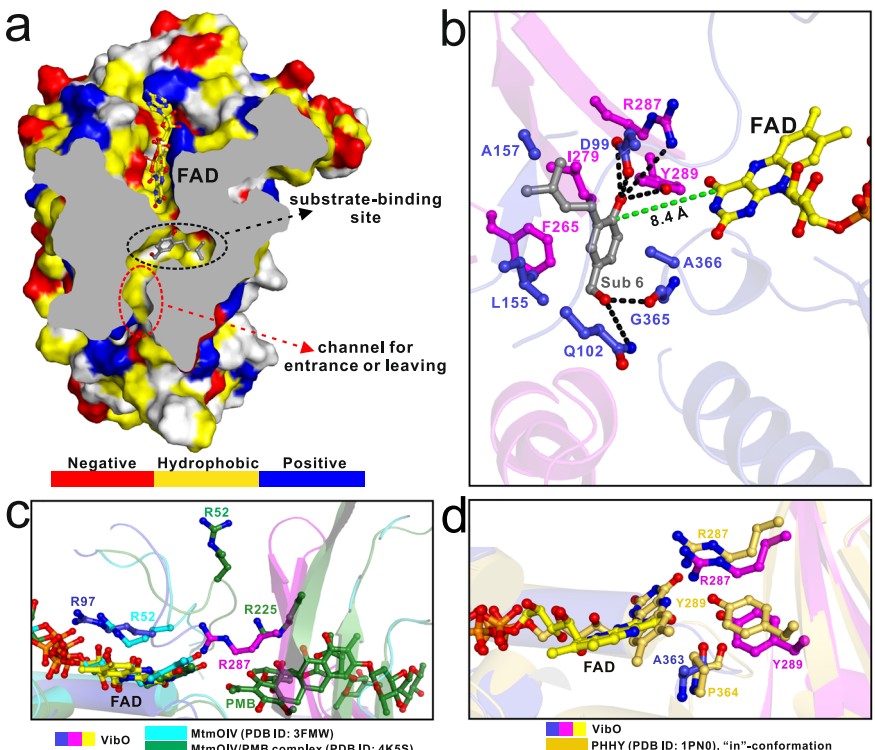

**Fig. 4 | Structural analyses of the active site of VibO. a** The combined surface representation and the stick-ball model showing the substrate-binding pocket docked with substrate **6** as well as the proposed substrate entrance or product leaving channel of VibO. **b** The ribbon-stick-ball representation showing the detailed interactions of substrate **6** (Sub 6) with the active site residues within the substrate-binding pocket of VibO in the structure model of VibO docked with substrate **6** by Autodock Vina program. The relevant hydrogen bonds involved in the interaction of substrate **6** with VibO in the complex model are shown as dotted black lines, and the distance between the carbon C6 of substrate **6** and the C4a atom of the bound FAD of VibO is further indicated and shown as a dotted green line. **c** The ribbon-stick-ball representation showing the detailed comparison of the proposed key R52 and R225 residues of the type-O BVMO MtmOIV for stabilizing the negatively charged flavin-OO$^-$ intermediate with the corresponding residues of VibO. **d** The ribbon-stick-ball representation showing the detailed comparison of the proposed key residues of the group A FMO PHHY for stabilizing the flavin-OOH intermediate with the corresponding residues of VibO. These structural analyses reveal that VibO contains essential structural elements for stabilizing both types of flavin co-factors (flavin-OO$^-$ and flavin-OOH).

is unusual for the group A flavin-dependent monooxygenases that largely require the substrate binding to trigger the flavin movement for access to NAD(P)H that reduces the flavin for priming the enzyme catalysis[36].

To probe the active flavin species, we analyzed spectral features and responses of VibO with NADPH in the absence of substrate **6**. The purified VibO showed the FAD spectrum ($\lambda_{max}$ = 450 nm), unlike the stable flavin-N5-oxide spectrum ($\lambda_{max}$ = 460 nm) observed in EncM[37] (Fig. 5b, Supplementary Fig. 36). In addition, more than 1 mM NADPH caused a rapid fall in the absorption of flavin in VibO (Fig. 5b). To further examine flavin changes of VibO with NADPH, we conducted time-course spectral detection and especially focused on absorption changes at 450 nm and 460 nm (Supplementary Fig. 36). Crucially, equal absorption at the beginning and the ending time points were observed for both wavelengths as the level of NADPH decreased over time (Fig. 5c), indicative of a complete reoxidation of the NADPH-reduced flavin. These spectrophotometric measurements may rule out the possibility of flavin-N5-oxide species which is stable in the absence of the substrate and has ~20% less absorption than flavin (Supplementary Fig. 36)[37], and support VibO involvement of the labile flavin-C4a-OO(H) which can be fully returned to flavin as observed (Fig. 5c).

To detect a possible hydroxy-oxepinone intermediate identified in the multifunctional GilOII that catalyzes successive hydroxylation and Baeyer-Villiger oxidation (Supplementary Fig. 23)[21], we conducted time-course analysis of VibO reactions quenched with methanol because a hydroxy-oxepinone as hemiketal/hemiacetal can be captured by methanol and the methanolyzed products were observed in reactions of FlsO1[25] and TerC[27]. The results showed that in all the

reactions with NADPH (0.1 to 5 mM), however, neither the putative hydroxy-oxepinone nor its methoxy derivatives could be detected by LC-MS (Supplementary Fig. 37). Considering that the phenol hydroxylase PHHY is structurally like VibO, we synthesized the PHHY sequence[29], expressed it in *E. coli*, and conducted enzyme assays using phenol or **6** as a substrate. While significant production of catechol was observed for PHHY incubating with phenol, neither **3** nor **7** was detected in the reaction with **6** (Supplementary Fig. 38).

Taken together, we propose two routes for the VibO catalysis (Fig. 5d, e). Following the PHHY mechanism[35,38], the hydroxylation logic to **7** as a side-product by VibO from **6** may involve an electrophilic attack on the phenol ring at its *ortho* position via hydroperoxyflavin FAD-C4a-OOH, resulting in the presumed dienone product **7'** which is converted spontaneously into the product **7** via keto-enol tautomerization (Fig. 5e). Given the structural similarity of VibO with PHHY, we speculated that VibO may adopt a similar approach to stabilize the FAD-OOH intermediate as PHHY (Fig. 4d). Since more NADPH led to the higher production of **7** yet a marked reduction in the yield of **3** where no conversion of **3** to **7** did occur (Fig. 2c, Supplementary Figs. 11 and 12), **3** and **7** are unlikely to share the common intermediate **7'**. Moreover, **7** cannot be transformed into **3** (Fig. 2a). Thus, a hydroxylation route to **3** via the intermediate **7'** would be refuted. Although it is unusual for a flavin-dependent monooxygenase to conduct both hydroxylation and Baeyer-Villiger oxidation, two reactive forms of flavin have been proposed to perform hydroxylation (flavin-OOH) or Baeyer-Villiger oxidation (flavin-OO$^-$) in the mechanism (Supplementary Fig. 23) for FlsO1 (fluoxanthone A)[25], LgnC (legonmycin)[39], and Rif-Orf17 (rifamycin)[40]. Since similar oxepinone formation occurs in the

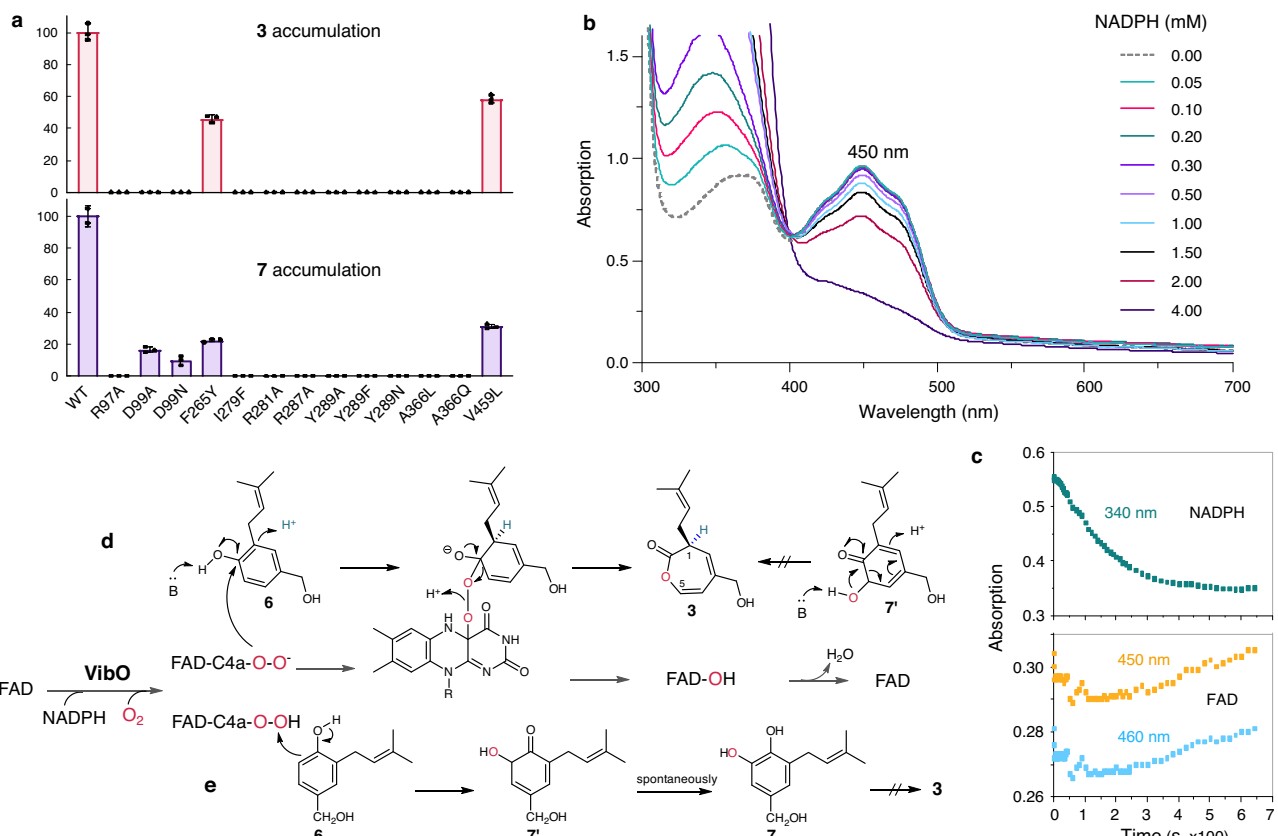

**Fig. 5 | Mutational and spectral analyses and the proposed VibO mechanism.**
**a** The relative enzyme activities for wild-type (WT) and variants of VibO incubated with **6**. The columns represent accumulation of **3** or **7** or none of them (in blank); the highest mean value was set 100%; bars indicate ± SD of three replicates.
**b** Reductive reaction of VibO with NADPH (0 to 4 mM) in the absence of substrate **6**. The gray dash denotes the purified VibO. Spectral responses were collected immediately after adding NADPH in a cuvette. **c** Time-course spectral analysis of

VibO with NADPH (50 μM) in the absence of **6** (Supplementary Fig. 36). The level of NADPH and FAD over time was monitored at 340 nm and 450 nm/460 nm, respectively. **d** Suggested peroxyflavin (FAD-C4a-O-O⁻)-involving Baeyer-Villiger oxidation of **6** to the oxepinone **3**. Deuterium incorporation is shown in blue. **e** Hydroxylation of **6** to **7** via the presumed dienone form **7'**, referring to the mechanism of PHHY[35,38]. Source data for **a**, **b**, and **c** are provided as a Source Data file.

catalysis of GilOII[21] and MtmOIV[24] (Supplementary Fig. 23), a Baeyer-Villiger oxidation via nucleophilic FAD-C4a-O-O⁻ attack to form the oxepinone in **3** seems plausible for the VibO catalysis where deuterium incorporation in **3** was obvious (Fig. 2d) and thus deprotonation of substrate **6** can be expected (Fig. 5d). Moreover, VibO has the structural elements potentially important for stabilizing FAD-OO⁻ (Fig. 4c). Structural analyses demonstrate that the active site residues and substrate-binding specificities of VibO are quite different from currently known fungal and bacterial Baeyer-Villiger oxygenases (Supplementary Fig. 39). While successive hydroxylation and Baeyer-Villiger oxidation by GilOII give rise to a hydroxy-oxepinone intermediate highly prone to decarboxylation for the gilvocarcin pathway[21], VibO, instead, may conduct separate oxygenation on **6** to give the oxepinone **3** via a Baeyer-Villiger oxidation and the catechol-containing **7** via a hydroxylation.

**Verification of 4 and 5 as intermediates en route to vibralactone in *B. vibrans***

To complete the vibralactone biosynthesis, we investigated the conversion of 4-hydroxybenzoate (**2**) to 3-prenyl-4-hydroxybenzylalcohol (**6**). As a known biosynthetic precursor 4-hydroxybenzoate is incorporated into ubiquinones and other meroterpenoids (e.g., asperpentyn[41], biscognienyne B[42]), starting with prenylation on its aryl ring. Vibralactone has a dimethylallyl group and its bicyclic lactone core can be derived from **2**, thus we hypothesized that the vibralactone biosynthesis may begin with prenylation of **2** to give 3-prenyl 4-

hydroxybenzoate (**4**), which may undergo stepwise reductions via 3-prenyl-4-hydroxybenzaldehyde (**5**) to the alcohol **6**. To test the hypothesis, we conducted the intermediate analog feedings as reported before (Supplementary Fig. 1)[15]. Our previous feeding [U-¹³C]-**2** led to the isolation of **3** (~30% labeled) and **1** (~43% labeled)[14], and feeding **6a** led to the purification of **3a** and **1a**[15], which demonstrated the conversion of **2** to **6** for the vibralactone (**1**) pathway. So, in this study to determine the intermediates between **2** and **6**, we use ESI−high-resolution MS (HRMS) with assistance of authentic standards (**6a**, **3a**, and **1a**) for analyzing **4a** and **5a** feedings. Compounds **4a** and **5a** (Fig. 6a), analogs of **4** and **5** that contain an allyl instead of a dimethylallyl group, were synthesized and fed (1 mM) respectively, to the growing cultures of *B. vibrans* once the accumulation of metabolite **3** can be observed by LC-MS, taking the broth immediately after feeding as control. The culture was incubated for additional 3 to 7 days, then the culture broth was extracted for LC-MS analyses and compared directly with the synthetic reference **6a** and authentic products (**1a**, **3a**, **3a'**) obtained from **6a**-feeding cultures in our work before[15]. Production of sodium adduct ions $[M+Na]^+$ at $m/z$ 187 for **6a**, $m/z$ 203 for **1a**, **3a**, and **3a'** (an isomer of **3a**) were exclusively revealed in both **4a**- and **5a**-feeding assays, and all the allyl metabolites were absent in controls (Fig. 6b). This was further supported by HRMS analyses, revealing product ions at $m/z$ 187.0731 as **6a** (calcd for $C_{10}H_{12}NaO_2^+$, $[M+Na]^+$: 187.0730), $m/z$ 203.0680 as **3a**, $m/z$ 203.0678 as **3a'** and **1a** (calcd for $C_{10}H_{12}NaO_3^+$, $[M+Na]^+$: 203.0679) (Fig. 6c, Supplementary Figs. 40–42). Thus, feeding experiments demonstrate that **4** and **5** are

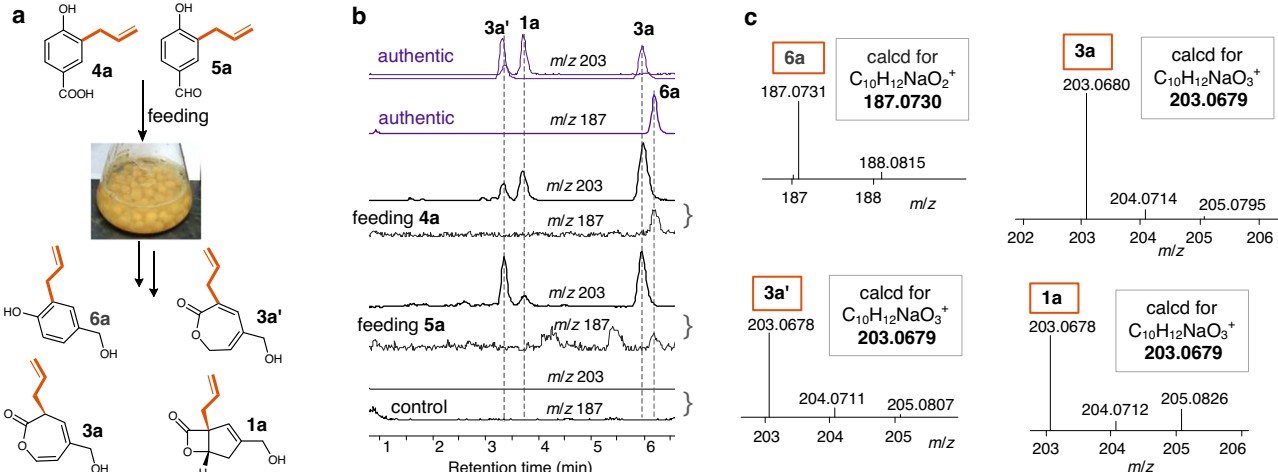

**Fig. 6 | In vivo transformation of analog intermediates in _B. vibrans_ mycelial cultures. a** Schematic illustration of feedings; **3a'** is an oxepin-2(7*H*)-one isomer of **3a**. **b** LC-MS analyses of the allyl metabolites produced in _B. vibrans_ mycelial feedings with 1 mM of **4a** or **5a**, showing extracted *m/z* chromatograms. **c** ESI-HRMS data for the allyl metabolites in (**b**). The original data are provided in Supplementary Figs. 40 and 41.

on-pathway biosynthetic intermediates leading to vibralactone (**1**) in _B. vibrans_.

## VibP1 and VibP2 function as membrane-bound UbiA prenyltransferases

Prenyltransferases that install dimethylallyl pyrophosphate (DMAPP) to 4-hydroxybenzoate (**2**) were recently reported in the biosynthesis of fungal meroterpenoids asperpentyn[41] and biscognienyne[42], and classified into the UbiA prenyltransferase family. We cloned the UbiA homologous sequences from mRNA of _B. vibrans_ and expressed in _E. coli_ BL21(DE3) (Supplementary Table 4). As UbiA prenyltransferases are known to be membrane bound, we prepared membrane extracts from the induced _E. coli_ and stored them at −80 °C for functional characterization.

To identify the enzymatic products from incubation with DMAPP and 4-hydroxyphenyl substrates, especially the site for prenylation on the phenol ring, we synthesized **4**, **5**, **6** and their C4-*O*-dimethylallyl congeners **4'**, **5'**, **6'** as authentic references (Supplementary Method 4 and Figs. 57–64). Reaction mixtures were analyzed by LC-MS and compared with the synthetic references. Out of eight candidates assayed, two proteins were shown to generate **4** as the only product from **2**, and no existence of **4'** was observed in the reactions. Production of **5** rather than **5'** from 4-hydroxybenzaldehyde, and **6** rather than **6'** from 4-hydroxybenzylalcohol were also evident from LC-MS analyses (Fig. 7a, b; Supplementary Figs. 43–45). Further ESI-HRMS found the product ions at *m/z* 205.0870 (calcd for $C_{12}H_{13}O_3^-$, [*M*−H]−: 205.0870) as **4**, *m/z* 189.0924 (calcd for $C_{12}H_{13}O_2^-$, [*M*−H]−: 189.0921) as **5**, *m/z* 191.1077 (calcd for $C_{12}H_{15}O_2^-$, [*M*−H]−: 191.1078) as **6** (Fig. 7d, Supplementary Figs. 46–48). The two prenyltransferase sequences, coming from different scaffolds of the genome and sharing 68% amino acid identity, are named as VibP1 and VibP2 thereafter. Their homologs *Sh*PT1/*Sh*PT2 (accession nos. XP_007303462 and XP_007299647) amplified from cDNA of _Stereum hirsutum_ can also accept the above 4-hydroxyphenyl substrates and form C3-prenyl products (Fig. 7a, Supplementary Figs. 43–45). To examine enzymatic activities with different 4-hydroxyphenyl substrates, the yield of **4**, **5**, and **6** was quantified on account of the standard courves. Results revealed that all the four prenyltransferases exhibited highest activity to 4-hydroxybenzoate (**2**) (Fig. 7a-c, Source Data file). When incubating with **2** and a mixture of four prenyl pyrophosphates including DMAPP ($C_5$), geranyl pyrophosphate (GPP, $C_{10}$), farnesyl pyrophosphate (FPP, $C_{15}$), and geranylgeranyl pyrophosphate (GGPP, $C_{20}$), VibP1 was shown to form **4** as a major product in addition to trace amounts of $C_{10}$ and

$C_{15}$ prenylated products (Supplementary Figs. 49 and 50). However, the soluble aromatic prenyltransferase formerly reported in _B. vibrans_ gave **6'** (C4-*O*-prenylation) as a major product and **6** (C3-prenylation) as a minor product when incubating with 4-hydroxybenzyl alcohol and DMAPP[14,43]. It is closer (67% sequence identity) to the basidiomycete BypB[44] (also soluble) for prenylation of orsellinic acid and has very low sequence identity (∼6%) with VibP1 and VibP2 (Supplementary Fig. 51), which are integral membrane proteins and clearly prefer 4-hydroxybenzoate (**2**) as substrate for C3-prenylation. These results demonstrate that VibP1/VibP2 catalyzes C3-prenylation on the 4-hydroxyphenyl ring of **2** to afford 3-prenyl 4-hydroxybenzoate (**4**), which is validated as an intermediate en route to vibralactone (**1**) in _B. vibrans_ (Fig. 6, Supplementary Figs. 40–42). Collectively, VibP1 and VibP2 function as UbiA prenyltransferases and are most likely involved in the vibralactone biosynthesis.

## Reductases catalyzing 4 to form 6

Since the reduction of carboxylic acids to the corresponding aldehydes and benzylic alcohols is well precedented in microbial systems, we set out to identify the relevant reductases by homology-based cloning from _B. vibrans_ mRNA with primers (Supplementary Tables 5 and 6) and functional characterization using recombinant proteins obtained from overexpression in _E. coli_ BL21(DE3). We amplified eight homologous sequences of the carboxylic acid reductase (CAR); among them only one protein, named as _Bv_CAR, was shown to catalyze reduction of 3-prenyl 4-hydroxybenzoate (**4**) to 3-prenyl-4-hydroxybenzaldehyde (**5**). The purified _Bv_CAR can give **5** as a sole product when incubating with **4** in the presence of both ATP and NAD(P)H. Higher relative activity was observed for _Bv_CAR incubating with NADPH in comparison to NADH, and no products can be observed without the exogenous ATP (Fig. 8a, b, Supplementary Figs. 52 and 53). The enzyme cannot reduce 4-hydroxybenzoate (**2**) to form 4-hydroxybenzaldehyde, indicating that _Bv_CAR is specific for production of **5** from **4** and most likely dedicated to the vibralactone pathway. Meanwhile, three aldehyde reductases (AR)/alcohol dehydrogenases (ADH), sharing a sequence identity of 22%∼68%, were identified to produce 3-prenyl-4-hydroxybenzylalcohol (**6**) from **5** in the presence of NAD(P)H and thereby named as _Bv_AR1, _Bv_AR2, and _Bv_AR3 individually. Much higher activity was observed for _Bv_AR1 incubating with NADPH than NADH (Fig. 8b, c, Supplementary Figs. 54 and 55). Considering reductases/dehydrogenases with relaxed substrate specificity, multiple enzymes including _Bv_ARs1–3 are likely to be involved in the reduction of **5** to **6** for the vibralactone biosynthesis.

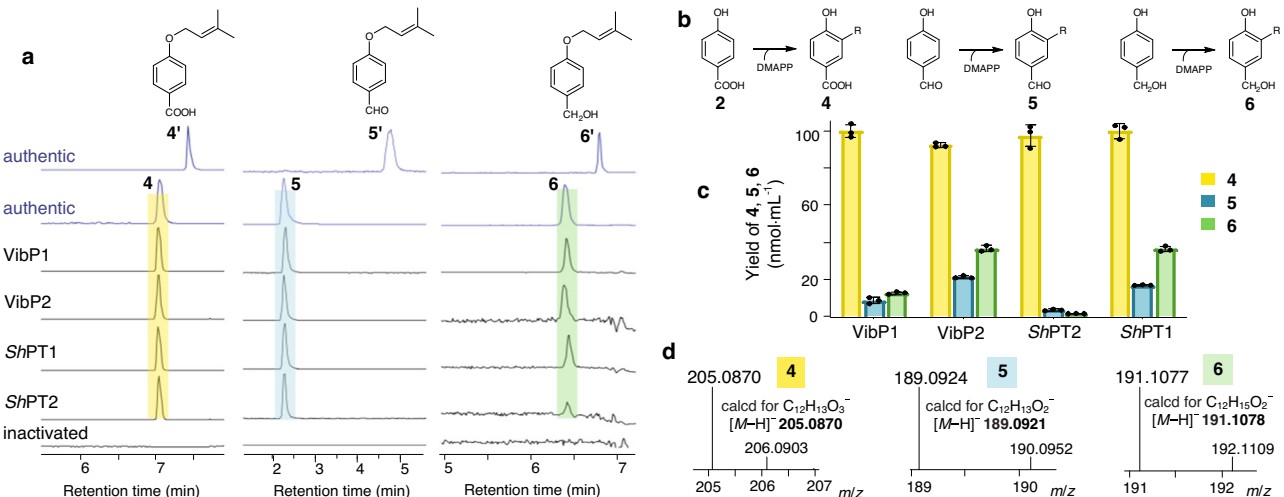

**Fig. 7 | Biochemical characterization of prenyltransferases expressed in *E. coli* as membrane proteins. a** LC-MS analyses of reactions illustrated in (**b**), showing extracted ion chromatograms (EIC) for the $[M-H]^-$ at $m/z$ 205 for **4** and **4'**, the $[M-H]^-$ at $m/z$ 189 for **5** and **5'**, the $[M+Na]^+$ at $m/z$ 215 for **6** and **6'**. Authentic references (**4, 4', 5, 5', 6'**) were synthesized in this study. **b** Reaction schemes of the prenyltransferases (R = dimethylallyl group). **c** The yield (nmol·mL⁻¹) of **4, 5**, and **6**

was quantified respectively by the peak area of EIC on account of the range of standard courves. The error bars indicate the standard deviation (SD). The data were generated by taking the mean values from three replicates and presented as mean values ± SD. Source data are provided as a Source Data file. **d** ESI-HRMS data for the reaction products (**4, 5, 6**) run in negative mode. The original data are provided in Supplementary Figs. 43–48.

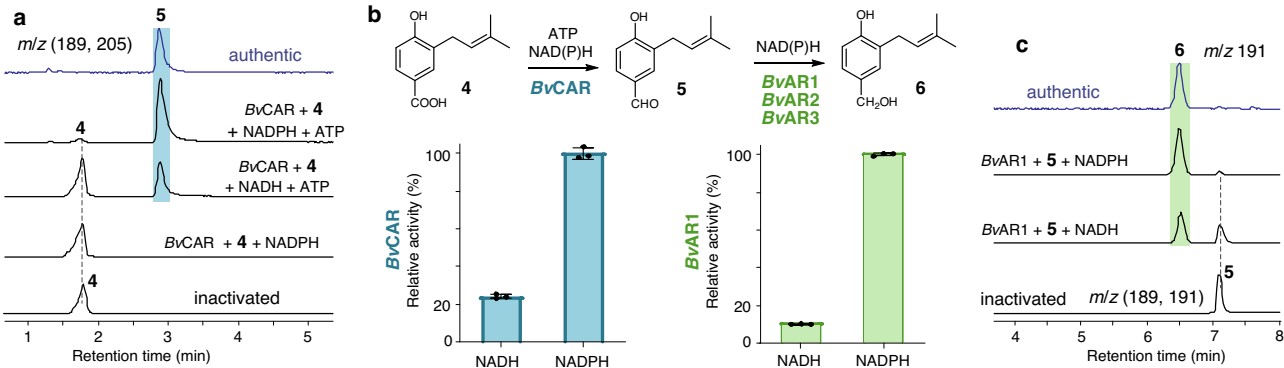

**Fig. 8 | Biochemical characterization of *Bv*CAR and *Bv*ARs expressed in *E. coli* as soluble proteins. a** LC-MS analyses of reactions with *Bv*CAR, showing extracted ion traces of −EIC (189, 205) overlay corresponding to the $[M-H]^-$ at $m/z$ 189 for **5** and $m/z$ 205 for **4**. **b** Reaction schemes and relative activities with NADH or NADPH. The columns represent accumulation of **5** (in blue) or **6** (in green) (the highest mean

value was set 100%); bars represent ± SD of three replicates. Source data are provided as a Source Data file. **c** LC-MS analyses of reactions with *Bv*AR1, showing extracted ion traces of −EIC (191) corresponding both to **6** $[M-H]^-$ and the natural-isotope ion **5**. The inactivated trace was extracted for −EIC (189, 191) overlay corresponding to **5**. The original data are provided in Supplementary Figs. 52 and 54.

## Reconstruction of the vibralactone pathway in vitro and in *E. coli*

After clarifying all previously unknown steps, we propose the biosynthetic pathway for vibralactone (**1**) as depicted in Fig. 1. To further validate the biosynthetic logic underlying vibralactone formation, we attempted to reconstruct the pathway both in vitro and in vivo. For in vitro reconstruction by means of one-pot as reported in the enzymatic total synthesis of enterocin[45] and defucogilvocarcin M[46], each of the five enzymes (VibP1, *Bv*CAR, *Bv*AR1, VibO, VibC) was overexpressed and purified from *E. coli* BL21(DE3). In the one-pot reaction mixture supplied with **2**, DMAPP, NADPH, and ATP, production of **1** and the oxepinone **3** was clearly observed by LC-MS analyses (Supplementary Fig. 56). For the complete pathway reconstructed in *E. coli* BL21(DE3), we co-expressed pBbA5C-MevT-MBIS[47] for overproduction of DMAPP and pET28a(+)-*VibP1-BvCAR-sfp-BvAR1-VibC-VibO* for the vibralactone pathway. Here a phosphopantetheinyl transferase (sfp) was expressed for activation of the CAR enzyme[48,49]. As a result, significant accumulation of **1** was observed in the *E. coli* transformants fed with **2**, and no detectable **1** existed in the control (Supplementary Fig. 56). Our efforts

on the potential of utilizing *E. coli* as a cell factory to produce **1** are still ongoing.

## Methods

### Fungal culture and feeding

Mycelia of the fungus *Boreostereum vibrans* from a seed culture on a plate for 7 days, was inoculated in 0.5 L modified PDB (potato 200.0 g, glucose 20.0 g, $KH_2PO_4$ 3.0 g, $MgSO_4$ 1.5 g, citric acid 0.1 g, and thiamin hydrochloride 10 mg in 1 liter of deionized water, pH 6.5), cultured at 25 °C on a rotary shaker at 140 rpm. Prior to mycelia harvest for enzyme fractionation, 0.1 mL of culture broth was removed from each bottle and extracted with equal volume of ethyl acetate three times. The dried extract was dissolved in 50 μL of methanol for LC-MS analyses to confirm the accumulation of 1,5-*seco*-vibralactone (**3**) in the culture. This procedure is essential to the subsequent enzyme fractionation because some of cultures have barely detectable **3** even from the same seed plate of inoculation. The mycelia (~18 days) significantly accumulating **3** were collected by filtration, washed with distilled

water, tapped with tissue paper to get rid of the remaining water, and frozen immediately in liquid nitrogen and stored at −80 °C until use.

For feeding experiments, the *B. vibrans* culture broth (0.2 mL) from each bottle was individually detected by LC-MS every day firstly on day 10 after inoculation. Once the accumulation of metabolite **3** can be observed by LC-MS, 1 mM of **4a** or **5a** were fed to the growing culture. The broth immediately after feeding was taken as control. The culture was incubated for additional 3–7 days, then the culture broth (0.2 mL) was extracted for LC-MS analyses. At least three replicates were conducted.

### Activity-guided fractionation and proteomic analysis
Proteins from *B. vibrans* mycelia were separated sequentially using ÄKTApure (GE Healthcare) at 7 °C by ion-exchange chromatography, hydrophobic interaction chromatography, and size-exclusion chromatography. The enriched enzyme fractions were subjected to LC-MS/MS for proteomic identification. See Supplementary Method 1 for details.

### Gene cloning and expression in *E. coli*
Total RNA was isolated from the *B. vibrans* mycelia significantly accumulating **3**, using the Plant RNA Mini Kit (Qiagen). The first-strand cDNA was synthesized with Superscript™ III First-strand Synthesis System (Invitrogen). The candidate genes were individually amplified by PCR using the corresponding primers (Supplementary Tables 2–6). The homolog sequences were obtained using RT-PCR from the cultured mushroom *Stereum hirsutum*. The resulting PCR products were purified and ligated into the vector pET28a(+) using a recombination cloning method. For site directed mutagenesis, rolling-cycle PCR amplification was performed using mutant primers (Supplementary Table 3) followed by DpnI digestion. Each mutation was confirmed by DNA sequencing. Recombinant plasmids were transferred into *Escherichia coli* BL21(DE3) for protein overexpression. The *E. coli* cells were induced with isopropyl-β-D-thiogalactopyranoside (IPTG, 0.1 mM for VibO and AR, 0.01 mM for CAR), followed by further incubation for 24 h at 16 °C (0.2 mM IPTG for 15 h at 25 °C for prenyltransferases). See Supplementary Method 2 for details. To obtain the membrane-bound prenyltransferases, the induced *E. coli* cells were suspended in 50 mM Tris-HCl buffer pH 7.5. The cleared lysate was further centrifuged at $150,000 \times g$ for 1 h at 4 °C to pellet the membrane fraction. Resuspended in the same buffer, the membrane fraction was assayed for enzyme activity or stored at −80 °C.

### In vitro assays of VibO
The standard enzyme (wild-type and variants) reaction was conducted in 100 μL of 50 mM sodium phosphate buffer pH 7.5 containing VibO (17 μM), the substrate (0.5 mM) and NADPH (0.1 mM); incubated at 28 °C for 2 h. Kinetic studies were performed with NADPH (0.1 mM) and varying concentrations of **6** (200 μM, 250 μM, 300 μM, 400 μM, 500 μM, 800 μM, 1500 μM), incubating at 28 °C for 30 min; all assays were conducted in triplicate. Prior to extraction with equal volume of ethyl acetate three times, 46 ng of 4-hydroxybenzophenone (Fluka) as an internal standard was added to each reaction mixture. The dried extract was dissolved in 50 μL of methanol and analyzed by LC-MS. The yield of enzyme-formed **3** was calculated from its peak area ratios to the internal standard, using the standard curve of authentic **3** (fungal culture-isolated) corresponding to its peak area ratios to the internal standard. Assays for variants were conducted in triplicate; the relative activities of variants were estimated from peak area ratios of products to their respective internal standards.

The reductive reaction of VibO (120 μM) with NADPH (0 to 4 mM), and the time-course (0 to 643 s) reaction of VibO (34 μM) with NADPH (50 μM) were carried out in a cuvette. The spectral data were collected immediately after additions of NADPH on a NanoDrop™ One UV-Vis Spectrophotometer (Thermo Scientific). The scan range was from 300 to 700 nm in 0.5 nm increments. The NADPH absorption was monitored at 340 nm; the flavin change was monitored at 450 nm and 460 nm.

### $^{18}$O-labeling reactions with $^{18}$O$_2$
The reaction mixture (2 mL in 50 mM sodium phosphate buffer pH 7.5) consists of 8 μM VibO and 0.1 mM NADPH into a round bottom flask (10 mL). The flask was purged with nitrogen for 30 min to thoroughly remove the atmospheric O$_2$. Then, $^{18}$O$_2$ (98% labeled) was introduced into the flask from the compressed gas cylinder via a syringe needle. Finally, 0.2 mM of **6** as substrate was added to the flask, incubating at 28 °C for 2 h. A scale-up reaction was conducted under $^{18}$O$_2$ to obtain enough labeled **3** for the $^{13}$C NMR analysis (Supplementary Fig. 15).

### Crystallization and structure determination
Details for VibO protein were described in Supplementary Method 3. Crystals of VibO were obtained using the sitting-drop vapor-diffusion method at 16 °C. The fresh purified VibO protein (10 or 20 mg·mL$^{-1}$ in 20 mM Tris-HCl, 100 mM NaCl, 1 mM DTT and 1 mM EDTA at pH 7.9) was mixed with equal volumes of reservoir solution containing 0.2 M ammonium acetate, 0.1 M Tris pH 8.0, 16% (w/v) polyethylene glycol 10000. Before diffraction experiments, 25% glycerol was added as the cryo-protectant. A 2.43 Å resolution X-ray data set for the native VibO in complex with FAD was collected at the beamline BL17B of the Shanghai Synchrotron Radiation Facility. The diffraction data was processed and scaled using XDS and autoPROC software suite[50–52]. The phase problem of VibO was solved by molecular replacement method using a predicted structural model by the Tencent T-fold sever as the search model with PHASER[53]. The initial structural models were rebuilt manually using COOT[54], and then refined using PHENIX[55]. Further manual model building and adjustments were completed using COOT. The qualities of the final models were validated by MolProbity[56]. The final refinement statistics of the solved structure in this study were listed in Supplementary Table 1.

### In vitro assays of PT, CAR, and AR
Reaction mixture for prenyltransferases (PT) contains 40 μL membrane extracts (from 6 mL of recombinant cells) in 50 mM Tris-HCl buffer (pH 7.5), 5 mM MgCl$_2$, 0.5 mM **2** or other substrates (50 mM stock in DMSO), 0.2 mM DMAPP, incubating at 30 °C for 3 h. For calculation of enzyme acitivies with different 4-hydroxyphenyl substrates, assays were conducted in triplicate. The yield of enzyme-formed products (**4, 5, 6**) was caculated by the peak area of EIC, on account of the range of standard courves. Reaction mixture for CAR contains 100 μL purified enzyme, 0.5 mM NAD(P)H, 1 mM ATP and 0.2 mM **4** or **2** as a substrate; reaction for AR contains 100 μL purified enzyme, 0.5 mM NAD(P)H, 0.2 mM **5** or 4-hydroxybenzaldehyde as a substrate. All incubate at 30 °C for 3 h. For calculation of enzyme acitivies with NADPH or NADH, assays were conducted in triplicate. Prior to extraction with equal volume of ethyl acetate, 46 ng of 4-hydroxybenzophenone as an internal standard was added in each reaction mixture. The relative enzyme activities of CAR and AR were estimated from peak area ratios of product to the internal standard.

### Pathway reconstruction by one-pot and in *E. coli*
The five-enzyme reaction mixture (200 μL in 50 mM Tris-HCl buffer pH 7.5) consists of freshly purified proteins (6 μM *Bv*CAR, 18 μM *Bv*AR1, 16 μM VibO, 4 μM VibC) and 90 μL membrane extracts of VibP1 (stored at −80 °C) and is supplied with 1 mM 4-hydroxybenzoate (**2**), 1 mM DMAPP, 1 mM NADPH, and 1 mM ATP, incubating at 30 °C for 3 h. The pathway reconstruction in *E. coli* was conducted with co-expression of pBbA5C-MevT-MBIS[47] for overproduction of DMAPP and pET28a(+)-*VibP1-BvCAR-sfp-BvAR1-VibC-VibO* for the vibralactone pathway (Supplementary Fig. 56). Plasmids were constructed by overlapping PCR (TransStart FastPfu DNA Polymerase) and Gibson Assembly (NovoRec

Plus PCR One Step PCR Cloning Kit) using primers listed in Supplementary Table 7. Since CAR must be activated into its holo form by a phosphopantetheinyl transferase (sfp), the *sfp* gene from *Bacillus subtilis* (accession WP_015715234.1) was synthesized and inserted into pET28a(+) for expression[48,49]. The *E. coli* BL21(DE3) transformants were fed with 2 mM **2** at 6 h after induction with 0.1 mM IPTG, followed by further incubation for 18 h at 30 °C. The culture broth was collected and extracted with ethyl acetate; then the dried extract was dissolved in 50 µL of methanol and analyzed by LC-MS.

### LC-MS analysis

The instrument Agilent 1290/6530 UPLC-Q-TOF conditions were optimized as Dual ESI: VCap 3500 V, Gas Temperature 350 °C, Drying gas 9 L·min⁻¹, Nebulizer 35 psig; MS TOF: Fragmentor 135 V, Skimmer 60 V, OCT1RFVpp 750 V. The mass spectrometer was run in positive or negative ionization mode and scanned from 50 to 500 *m/z*. For high-resolution MS, mass calibration was achieved by using hexakis(1*H*, 1*H*, 3*H*-tetrafluoropropoxy)phosphazine, purine and trifluoroacetic acid ammonium salt in each run. Samples (1.0 µL for each injection) were separated by an Agilent ZORBAX Eclips Plus C18 Rapid Resolution HD column (2.1 mm × 50 mm, 1.8 µm pore size) at a flow-rate of 0.3 mL·min⁻¹ (30 °C). Data were collected by Agilent MassHunter Workstation Data Acquisition and analyzed by Qualitative Analysis B.06.00-ZRR3.m. Authentic references were analyzed in parallel with samples. Preparative HPLC was run on a Waters 1525 system, using a XBridge Prep C18-OBD column (19.0 mm × 250 mm, 5.0 µm pore size). Chromatographic separation was described individually in supplementary figure legends. For the standard courves of synthesized references (**4**, **5**, **6**) and calculation of prenyltransferase acitivities with different 4-hydroxyphenyl substrates, the chromatographic separation was performed with elution of 43% B over 4.9 min and 100% B over the next 3.1 min where B was methanol, A was 0.1% formic acid for **4** or H₂O for **5** and **6**. The mass spectrometer was run in negative ionization mode.

### Chemical synthesis

Compounds including **4a**, **5a**, **4**, **4′**, **5**, **5′**, and **6′** were synthesized in this study. Their identity was verified by ¹H-NMR and ¹³C-NMR spectra (Supplementary Figs. 57–64). See Supplementary Method 4 for details.

### Reporting summary

Further information on research design is available in the Nature Portfolio Reporting Summary linked to this article.

## Data availability

The coordinate and structure factor of VibO in complex with FAD solved in this study have been deposited in the Protein Data Bank under the accession code 7YJ0. The data generated in this study are provided in the Supplementary Information/Source Data file. Sequences have been deposited in GenBank (https://www.ncbi.nlm.nih.gov) under accession nos. OP484926 (*VibO*), ON653009 (*VibP1*), ON653010 (*VibP2*), ON653011 (*BvCAR*), ON653012 (*BvAR1*), ON653013 (*BvAR2*), and ON653014 (*BvAR3*), respectively. Source data are provided with this paper.

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

## Acknowledgements

This work was supported by grants National Key R & D Program of China (2018YFA0900600 to Y.Z.), the National Natural Science Foundation of China (21977101 to Y.Z., 21961142008 to J.-K.L., 92253301 and 21822705 to L.P.), the Yunnan Key Research and Development Program (2019ZF011-2 to Y.Z.), the Science and Technology Commission of Shanghai Municipality (20XD1425200 to L.P.), CAS Special Research Assistant Project and Postdoctoral Directional Training Foundation of Yunnan Province (K.-N.F.). We thank SSRF BL17B and BL10U2 for X-ray beam time.

## Author contributions

K.-N. Feng and Y. Zhang conducted the biochemical, genetic, and chemical experiments. M. Zhang performed crystallization experiments. Y.-L. Yang assisted in chemistry. J.-K. Liu, L. Pan and Y. Zeng directed the research. L. Pan and Y. Zeng wrote the manuscript with input from all authors.

## Competing interests

The authors declare no competing interests.
