## [Peer Review File · Nature Communications]

REVIEWER COMMENTS

Reviewer #1 (Remarks to the Author):

This paper describes discovery and *in vitro*/reconstitution of an oxepinone producing pathway. Various enzymes (a prenyltransferase, carboxylic acid reductase and a flavin monooxygenase amongst others) are involved and have been studied. Most focus occurs on the flavin monooxygenase, for which some solution properties and an accompanying crystal structure is provided. These, according to the authors support a BV type oxidation mechanism adapted by VibO.

At present I do not fully agree with the abstract sentences:

The crystal structure of VibO reveals that it forms a dimeric phenol hydroxylase-like architecture featured with a unique substrate-binding pocket adjacent to the bound FAD. Importantly, relevant biochemical and

structural analyses support a Baeyer-Villiger oxidation mechanism adopted by VibO for catalyzing the oxepinone formation.

It is my opinion that the biochemical and structural studies on VibO do not provide much information on mechanism, and Fig 5 does not rely on structural insights into the active site. The active site is ill defined at present, as the structure is of the "flavin-out state" and relies heavily on similarity with a phenol hydroxylase (which catalyses a reaction similar to the VibO side reaction). In the case of Fig 5a, the notion that protonation (and loss of aromaticity) occurs before formation of the putative C4a-OO adduct is without any evidence. Similarly, the presence of such an C4a adduct is without direct evidence, but merely on the basis of similarities with other enzymes. Recent developments in flavin enzymology now also highlight the existence of N5-oxidative species. While I think this is a good paper, the hypothesis in Fig 5 remains largely hypothetical in absence of further studies. This might well be outside the scope of this paper, but could include:

making use of the flavin and NADPH spectral properties to provide more insight into mechanism, using the spectral responses of flavin to reduction/change in environment through binding of substrate(s). Similarly, NADPH consumption/oxidation can be directly monitored at 340nm and can link the stoichiometry of the reaction. Furthermore, the effect of mutation on FAD binding/incorporation should be verified using spectral analysis.

The authors highlight they attempted to get a complex crystal structure, but do not submit the data/structure to the PDB. Fig S24 reveals some density (although 1 sigma for FoFc is very low) for the complex, and this should either be refined/submitted or removed from the manuscript, possibly replaced by modelling of the substrate complex. In this regard, the flavin in/out conformation will be relevant to substrate binding. A plausible model of the michaelis menten complex might provide more insight supporting Fig 5.

As such, the notion the VibO active site is unique is ill-defined here, and the fact that biochem/structure support the BV proposal is tentative.

Minor comments.

Line 478, apo-form usually refers to no-FAD?

Methods: microsome extracts usually refers to eukaryotic preparations, so E. coli membrane preparation might be better?

Table S1, CC1/2 should ideally be provided

Some figure with electron density for active site/flavin should be shown.

Reviewer #2 (Remarks to the Author):

This is a good manuscript, with a lot of data supporting most of the authors' hypotheses, even though some of these are rather obvious, like the reduction steps from 4 to 6. The central focus of the manuscript is the Baeyer-Villiger oxygenase VibO, albeit this is also obvious from structure 3. The reconstitution of the pathway is nice work, but several reconstitution pathways, some including many more enzymes, were published before, e.g., the pathway to defuco-gilvocarcin or enterocin, and not cited.

The introduction section needs to be overhauled, there are grammar errors, awkward or unclear sentences.

The crystal structure is good, but lacks follow up data and certain discussions, e.g., there is no discussion about a peroxyflavin stabilizing arginine residue that is typical for BV oxygenases. How can the enzyme stabilize both types of flavin co-factors (with OO- or OOH)?

It is also unfortunate that experiments failed to co-crystallize the putative substrate with the enzyme. How about soaking experiments? How about enzyme-product co-crystallization? This and the lack of site-directed mutagenesis mutants along with kinetic data makes all conclusions regarding the substrate binding pocket speculative, thus renders the protein crystal structure less useful than what you would expect in context with these enzyme studies.

The fact that VibO can catalyze both a BV reaction and a phenol hydroxylation is intriguing but seems unlikely. At least it should be discussed in context with the crystal structure how the enzyme might stabilize both, flavin-OO- (peroxyflavin) and a flavin-OOH? Apparently, this is due to NADPH concentrations, and the phenol hydroxylation is favored as long as sufficient reduction power in form of NADPH is available. Thus, it might also be possible that both reactions can occur subsequently (like shown for GilOII), and the real product wasn't detected, possibly because it would lead to an unstable aldehyde (via a hydroxy-oxepinone intermediate) containing product that disappears due to its reactivity, see suppl. Fig. 12! The crystal structure, most similar to PHHY, also supports that thought. If so, then both observed products 3 and 7 may be shunt pathways reflecting leaking or substrate-flexible enzymes, in context with available NADPH concentrations.

Mechanistically, the VibO BV-reaction is not that different from MtmOIV, XanO4 or GilOII. The decarboxylation reaction, mentioned as one of the differences, following the latter three enzymes is aided by a beta-carbonyl, which is not found in the vibrilactone biosynthetic intermediates, thus cannot happen for VibO due to the substrate structure, rather than the enzyme itself.

Some minor comments:

Suppl. Fig 4 seems questionable. Substrate and product apparently have the same retention time in the shown HPLC traces.

Fig. 7: what is the peak at ~ 7.2 min?

Overall, this is a thoroughly conducted study leading to a data-rich manuscript, but this seems to be thrown together with data from various experiments, lacking depth in the key topic (VibO studies). As is, the manuscript might lack the originality and depth required for a top journal as Nature Communications. It might be better to finish the VibO studies and publish these alone, while the reconstitution pathway and other studies could be published elsewhere.

Reviewer #3 (Remarks to the Author):

387155 Pan, Zeng et al

Recommendation: suitable for publication in nature communications with minor edits.

The authors present the complete biosynthesis of the fungal metabolite vibrallactone. They isolate, express, and characterize key enzymes for prenylation and oxepinone formation, and reconstruct the pathway heterologously.

This study combines a tremendous amount of work that leads to the production of vibrallactone. This biologically active beta lactone is produced by *Boreostereum vibrans*, a basidiomycete fungus, for which biosynthetic gene clusters are unlikely clustered and genome mining approaches are very limited.

Here, the team succeeded in purifying a Baeyer-Villiger oxygenase VibO, a key enzyme in the vibrallactone biosynthesis which forms an oxepine intermediate, via enzyme- activity guided fractionation. They carefully biochemically characterize the unique enzyme VibO, obtain a crystal structure, create site mutants, and perform isotope feeding studies. Additionally, the prenyltransferases VibP1 and VibP2 as well as the reductase BvCAR are characterized and expressed in *e.coli*.

Finally, a combination of the five key biosynthetic enzymes in a one-pot reaction as well as biosynthetic pathway engineering in vivo in *e.coli* are successfully conducted to yield vibrallactone. What a comprehensive study - Congratulations to the authors!

While the manuscript is generally clear and experiments well thought through, there are a few minor comments and questions that could improve the manuscript:

- Baeyer-Villiger oxygenase VibO represents a new enzyme which should be put into context of other fungal Baeyer-Villiger oxygenases like the aflatoxin oxygenase or others with regards to their crystal structure, active site amino acids, substrate specificity.
- The isotope feeding to verify the activity of VibO, figure 2D, should include MS/MS data to further verify the isotope location within the metabolite fragments. It is not quite clear if the isotope analysis is based on MS only or if the labelled compounds were isolated and characterized? ¹⁸O and D labels

would be nicely visible in ^{13}C or ^1H spectra. The caption for figure 2 should list supplemental figures 15 and 16 for 2D.

- Feeding experiment Figure 6: 1mM 4a and 5a were added to the fungal cultures, what was the yield? Production analysis is based on HRMS only, quantification of metabolites should be added or explained why EIC was used only.
- The optical rotation was recorded at different concentrations for isolated 3 and enzyme produced 3. This should have been done at the same concentration.
- The estimated total yields of 3 from in vitro and in vivo production should be listed to give context to the fungal, WT production.

Minor comments:

- Supplement figure 1 states 'dozens of milligrams' please give a number. The chromatographic separation is quite unusual: Chromatographic separation of 6a, 3a, and 3a' (an isomer of 3a) was conducted with elution of 25% B over 4.8 min and 100% B over the next 3.2 min where A was H₂O and B was methanol. The mass spectrometer was run in positive ionization mode. Can the authors comment on why not a longer HPLC run was used or a flatter gradient was tried? In Supplement figure 4, the retention time of 6 and 3 is almost overlapping, while for other chromatograms (Supplemental figure 5), the separation is ok, although the same experimental settings are listed. In many figures, and for certain compounds, negative ionization mode was used.
- Supplemental figure 7: dried for 'two overnights' please give precise time information.
- Supplemental figure 8: caption states that 70 mg of 6 was converted, while the main text states 90mg?
- Supplement figure 16 does not show lines for some of the base ions.
- Supplement figure 19: I am not an expert in sedimentation coefficients, but a standard protein with known MW and resulting S value might be needed for context?

Point-by-point responses to the reviewers' comments:

(Reviewers' comments are in **blue**, and our responses are in **black**)

Specific responses to the criticisms from the Reviewer #1:

This paper describes discovery and *in vitro*/reconstitution of an oxepinone producing pathway. Various enzymes (a prenyltransferase, carboxylic acid reductase and a flavin monooxygenase amongst others) are involved and have been studied. Most focus occurs on the flavin monooxygenase, for which some solution properties and an accompanying crystal structure is provided. These, according to the authors support a BV type oxidation mechanism adapted by VibO.

At present I do not fully agree with the abstract sentences:

The crystal structure of VibO reveals that it forms a dimeric phenol hydroxylase-like architecture featured with a unique substrate-binding pocket adjacent to the bound FAD. Importantly, relevant biochemical and structural analyses support a Baeyer-Villiger oxidation mechanism adopted by VibO for catalyzing the oxepinone formation.

Following reviewer's comments, we have revised the **abstract**.

It is my opinion that the biochemical and structural studies on VibO do not provide much information on mechanism, and Fig 5 does not rely on structural insights into the active site. The active site is ill defined at present, as the structure is of the "flavin-out state" and relies heavily on similarity with a phenol hydroxylase (which catalyses a reaction similar to the VibO side reaction). In the case of Fig 5a, the notion that protonation (and loss of aromaticity) occurs before formation of the putative C4a-OO adduct is without any evidence. Similarly, the presence of such an C4a adduct is without direct evidence, but merely on the basis of similarities with other enzymes. Recent developments in flavin enzymology now also highlight the existence of N5-oxidative species. While I think this is a good paper, the hypothesis in Fig 5 remains largely hypothetical in absence of further studies. This might well be outside the scope of this paper, but could include:

making use of the flavin and NADPH spectral properties to provide more insight into mechanism, using the spectral responses of flavin to reduction/change in environment through binding of substrate(s). Similarly, NADPH consumption/oxidation can be directly monitored at 340nm and can link the stoichiometry of the reaction. Furthermore, the effect of mutation on FAD binding/incorporation should be verified using spectral analysis.

We thank the reviewer for the above constructive comments and suggestions. To probe the active flavin species, we analyzed spectral features and responses of VibO with NADPH in the absence of the substrate **6**. The purified VibO showed the FAD spectrum ($\lambda_{\max}=450$ nm), unlike the stable flavin-N5-oxide spectrum ($\lambda_{\max}=460$ nm) observed in EncM (see below, **Fig. 5b**, **Supplementary Fig. 36**). In addition, more than 1 mM NADPH caused a rapid fall in the absorption of flavin in VibO (**Fig. 5b**). To further examine flavin changes of VibO with NADPH, we conducted time-course spectral detection and especially focused on absorption changes at 450 nm and 460 nm (**Supplementary Fig. 36**). Crucially, equal absorption at the beginning and the ending time points were observed for both wavelengths, indicative of a complete reoxidation of the NADPH-reduced flavin (**Fig. 5c**). These spectrophotometric measurements may rule out the possibility of flavin-N5-oxide species which is stable in the absence of the substrate and has ~20% less absorption than flavin (Supplementary Fig. 36), and support VibO involvement of the labile flavin-C4a-OO(H) which can be fully returned to flavin as observed (Fig. 5c).

Following reviewer's suggestions, we performed spectral analysis of the effect of mutation on the substrate binding, FAD incorporation and NADPH consumption, which presented in the revised manuscript including **Figure 5b-c** and **Supplementary Fig. 32-36**.

The authors highlight they attempted to get a complex crystal structure, but do not submit the data/structure to the PDB. Fig S24 reveals some density (although 1 sigma for FoFc is very low) for the complex, and this should either be refined/submitted or removed from the manuscript, possibly replaced by modelling of the substrate complex. In this regard, the flavin in/out conformation will be relevant to substrate binding. A plausible model of the michaelis menten complex might provide more insight supporting Fig 5. As such, the notion the VibO active site is unique is ill-defined here, and the fact that biochem/structure support the BV proposal is tentative.

We understand the reviewer's concern, and thank the reviewer for the above constructive comments. Following the reviewer's suggestion, we have removed the original Fig S24, and conducted a structural modelling analysis of VibO with substrate **6** using the Autodock Vina program. A search for the lowest binding energy and the best complementary shape led to a reasonable conformer of substrate **6** (see **Fig. 1a** below, and **Fig. 4a** in the revised manuscript), which accommodates the cavity well and forms favorable hydrophobic and polar interactions with the surrounding VibO residues, including D99, L155, A157, F265, I279, Y289, and A366 (see **Fig. 1b** below, and **Fig. 4b** in the revised manuscript). Importantly, our site-directed mutagenesis-based analyses showed that point mutations of key active site residues of VibO, such as F265Y, I279F, Y289A, A366L, and A366Q, all dramatically decreased the enzymatic activity of VibO, further validating our structural modeling result. We have updated Fig.4 and included these docking results in the revised manuscript.

Fig. 1: Structural docking analyses of VibO with substrate 6. (a) The combined surface representation and the stick-ball model showing the substrate-binding pocket docked with the substrate 6 as well as the proposed substrate entrance or product leaving channel of VibO. (b) The ribbon-stick-ball representation showing the detailed interactions of substrate 6 (Sub 6) with the active site residues within the substrate-binding pocket of VibO in the structure model of VibO docked with substrate 6 by Autodock Vina program. The relevant hydrogen bonds involved in the interaction of substrate 6 with VibO in the complex model are shown as dotted black lines, and the distance between the carbon C6 of substrate 6 and the C4a atom of the bound FAD of VibO is further indicated and shown as a dotted green line.

Minor comments.

Line 478, apo-form usually refers to no-FAD?

We thank the reviewer for pointing out this inappropriate statement in our original manuscript. Following reviewer's comment, we have changed the "apo-form VibO" to "native VibO in complex with FAD" in the revised manuscript.

Methods: microsome extracts usually refers to eukaryotic preparations, so E. coli membrane preparation might be better?

Yes, corrected.

Table S1, CC1/2 should ideally be provided

Following reviewer's comments, we have included the CC1/2 values and updated the Table S1 in the revised manuscript.

Some figure with electron density for active site/flavin should be shown.

Following reviewer's comment, we have added electron density of flavin into the updated Supplementary Fig. 27 in the revised manuscript (see Fig. II below, and Supplementary Fig. 27 in the revised manuscript).

Fig. II: The molecular interface of VibO and FAD interaction. A stereo view of the ribbon-stick-ball model showing the detailed binding interface between VibO and the cofactor FAD in the structure of VibO in complex with FAD. The hydrogen bonds and salt bridges involved in the interactions are shown as dotted lines. The F_o-F_c map is shown as a cyan mesh, and is calculated from the final PDB file by omitting the FAD molecule and contoured at 2.5σ .

Specific responses to the criticisms from the Reviewer #2:

This is a good manuscript, with a lot of data supporting most of the authors' hypotheses, even though some of these are rather obvious, like the reduction steps from 4 to 6. The central focus of the manuscript is the Baeyer-Villiger oxygenase VibO, albeit this is also obvious from structure 3. The reconstitution of the pathway is nice work, but several reconstitution pathways, some including many more enzymes, were published before, e.g., the pathway to defuco-gilvocarcin or enterocin, and not cited.

Following reviewer's suggestions, we have cited the two publications in the revised manuscript, and listed below:

46. Cheng, Q. et al. Enzymatic total synthesis of enterocin polyketides. *Nat. Chem. Biol.* **3**, 557–558 (2007).

47. Pahari, P. et al. Enzymatic total synthesis of defucogilvocarcin M and its implications for gilvocarcin biosynthesis. *Angew. Chem. Int. Ed. Engl.* **51**, 1216–1220 (2012). *Angew. Chem.* **124**, 1242–1246 (2012).

The introduction section needs to be overhauled, there are grammar errors, awkward or unclear sentences.

We thank the reviewer for pointing out these. Following reviewer's comment, we have revised the **introduction section**.

The crystal structure is good, but lacks follow up data and certain discussions, e.g., there is no discussion about a peroxyflavin stabilizing arginine residue that is typical for BV oxygenases. How can the enzyme stabilize both types of flavin co-factors (with OO- or OOH)?

We thank the reviewer for this constructive comment. Following the reviewer's insightful suggestion, we have conducted careful structure comparison analyses of VibO with currently known flavin-dependent BV oxygenases. Currently, there are several distinct types of flavin-dependent monooxygenases (FDMO) that catalyze BV oxidations, including type-I BVMOs (belongs to Class B FDMOs), type-II BVMOs and atypical type-O BVMOs. So far, there are only five type-I BVMOs and one type-O BVMO (MtmOIV) with available structural information. Based on our structural analyses, the overall structure of VibO is similar to that of MtmOIV (a type-O BVMO) but distinct from that of type-I BVMOs, such as the phenylacetone monooxygenase PAMO from *Thermobifida fusca* and BVMO_{AFL838} from *Aspergillus flavus* (see **Fig. IIIa, b, f** below, and **Supplementary Fig. 25b, c** in the revised manuscript). Importantly, the critical Arg residue that

stabilizes the negatively charged flavin-OO⁻ intermediate in type-I BVMOs is missing in VibO (see **Fig. IIIc-e** below, and **Supplementary Fig. 39a-c** in the revised manuscript), while the proposed Arg residues (R52 and R225 in MtmOIV) that can stabilize the negatively charged flavin-OO⁻ intermediate in MtmOIV are well conserved in VibO (R97 and R287 in VibO) (see **Fig. IIIh** below, and **Fig. 4c** in the revised manuscript). Therefore, the way adopted by VibO to stabilize flavin-OO⁻ intermediate is likely to be similar to that of MtmOIV, but definitely, should be different from that of type-I BVMOs. Interestingly, the overall structure of VibO is very similar to that of group A FDMOs, such as the phenol hydroxylase PHHY (see **Fig. IIIg** below, and **Supplementary Fig. 25a** in the revised manuscript). Meanwhile, it is reported that the flavin-OOH intermediate is stabilized to some extent by nearby residues in group A FDMOs. For instance, PHHY can stabilize the flavin-OOH intermediate by a hydrogen-bond formed between the carbonyl oxygen of P364 residue and the hydrogen on the OOH moiety of the flavin hydroperoxide based on a quantum mechanical/molecular mechanical study of PHHY (1), or a hydrogen-bond formed between the side chain of PHHY Y289 residue and the hydrogen on the OOH moiety of the flavin hydroperoxide based on a previous structural modelling analysis (2). Nevertheless, detailed structural analyses showed that like PHHY, VibO contains a similar backbone carbonyl oxygen group of A363 and a Tyr residue at corresponding regions (see **Fig. IIIi** below, and **Fig. 4d** in the revised manuscript). Given the structural similarity of VibO with PHHY, we speculated that VibO might adopt a similar approach to stabilize the flavin-OOH intermediate as PHHY. Therefore, VibO is likely to have essential structural elements for stabilizing both types of flavin co-factors (flavin-OO⁻ and flavin-OOH). We have included these structure comparison analyses of VibO with type-I BVMOs, the type-O BVMO MtmOIV and the group A FDMO PHHY in the revised manuscript.

Fig. III: Structural comparison analyses of VibO with other relevant flavin-dependent monooxygenases. (a and b) The combined ribbon and stick representation showing the overall structure comparison of the monomeric VibO and the type-I BVMO PAMO (green, PDB ID: 2YLZ) (a), or BVMO_{AFL838} (pink, PDB ID: 5J7X) (b). (c and d) The ribbon-stick-ball representation showing the enlarged view of the key peroxyflavin stabilizing arginine residue of type-I BVMO PAMO (c), or BVMO_{AFL838} (d) above the flavin ring of the co-factor FAD. (e) The ribbon-stick-ball representation showing the structural comparison of the key peroxyflavin stabilizing arginine residues of type-I BVMO PAMO and BVMO_{AFL838} with the corresponding I337 residue of VibO. (f and g) The combined ribbon and stick representation showing the overall structure comparison of the monomeric VibO and the type-O BVMO MtmOIV (forest green, PDB ID: 4K5S) (f), or the Class A FDMO PHHY (orange, PDB ID: 1PN0) (g). (h) The ribbon-stick-ball representation showing the detailed comparison of the proposed key R52 and R225 residues of the type-O BVMO MtmOIV for stabilizing the negatively charged flavin-OO⁻ intermediate

with the corresponding residues of VibO. (i) The ribbon-stick-ball representation showing the detailed comparison of the proposed key residues of the Class A FDMO PHHY for stabilizing the flavin-OOH intermediate with the corresponding residues of VibO. These structural analyses reveal that VibO contains essential structural elements for stabilizing both types of flavin co-factors (flavin-OO⁻ and flavin-OOH).

It is also unfortunate that experiments failed to co-crystallize the putative substrate with the enzyme. How about soaking experiments? How about enzyme-product co-crystallization? This and the lack of site-directed mutagenesis mutants along with kinetic data makes all conclusions regarding the substrate binding pocket speculative, thus renders the protein crystal structure less useful than what you would expect in context with these enzyme studies.

We understand the reviewer's concern. In order to fully understand the substrate binding pocket of VibO, we need to obtain a relevant complex structure of VibO. Actually, we have tried many approaches to solve the complex structure of VibO either with the substrate or product, including co-crystallization and soaking experiments. But, unfortunately, despite numerous attempts, none of the crystals analyzed showed reasonable electron density for the bound substrate or product, likely due to the poor solubility of the substrate in the buffer solution as well as the weak binding between VibO and the product. Fortunately, as an alternative, we have used the docking program Autodock Vina to obtain a reasonable structure model of VibO in complex with the substrate (see **Fig. I** above, and **Fig. 4a,b** in the revised manuscript), which is well consistent with our biochemical results. Also see our detailed response to the major point from the Reviewer 1.

The fact that VibO can catalyze both a BV reaction and a phenol hydroxylation is intriguing but seems unlikely. At least it should be discussed in context with the crystal structure how the enzyme might stabilize both, flavin-OO⁻ (peroxyflavin) and a flavin-OOH?

Although it is unusual for a flavin-dependent monooxygenase to conduct both hydroxylation and Baeyer-Villiger (BV) oxidation, two reactive forms of flavin have been reported to perform BV-oxidation (via Fl-O-O⁻) and hydroxylation (via Fl-O-OH) in the mechanism of FlsO1, LgnC, and Rif-Orf17 (see below, **Supplementary Fig. 23**).

Apparently, this is due to NADPH concentrations, and the phenol hydroxylation is favored as long as sufficient reduction power in form of NADPH is available.

Yes.

Thus, it might also be possible that both reactions can occur subsequently (like shown for GilOII), and the real product wasn't detected, possibly because it would lead to an unstable aldehyde (via a hydroxy-oxepinone intermediate) containing product that disappears due to its reactivity, see suppl. Fig. 12!

We understand the reviewer's concern. To detect the putative hydroxy-oxepinone, we have conducted time-course reactions quenched with methanol since a hydroxy-oxepinone as hemiketal/hemiacetal can be captured by methanol (**Supplementary Fig. 23**). In all the reactions, however, neither the putative hydroxy-oxepinone nor its methoxy derivatives could be observed by LC-MS detection (**Supplementary Fig. 37**).

The crystal structure, most similar to PHHY, also supports that thought. If so, then both observed products **3** and **7** may be shunt pathways reflecting leaking or substrate-flexible enzymes, in context with available NADPH concentrations.

This study identified VibO as an oxepinone-forming enzyme for the vibrallactone pathway, based on (i) in vitro production of **3** and no detectable hydroxy-oxepinones, (ii) in vivo production of **3** by the *vibO*/pET28a/*E. coli* strain fed with **6**, (iii) in situ accumulation of **3** and its cyclization to vibrallactone (**1**) in the fermented *B. vibrans* fungi. Description of 'shunt' has been removed from the manuscript.

Mechanistically, the VibO BV-reaction is not that different from MtmOIV, XanO4 or GilOII. The decarboxylation reaction, mentioned as one of the differences, following the latter three enzymes is aided by a beta-carbonyl, which is not found in the vibrallactone biosynthetic intermediates, thus cannot happen for VibO due to the substrate structure, rather than the enzyme itself.

Although they may perform mechanistically similar hydroxylation and BV-oxidation, VibO and GilOII have distinct structure and substrate specificity. GilOII catalyzes successive hydroxylation, BV-oxidation, and decarboxylation for the gilvocarcin pathway. VibO, however, may perform separate oxygenation on **6** to give the oxepinone **3** and the hydroxylation product **7**.

Some minor comments:

Suppl. Fig 4 seems questionable. Substrate and product apparently have the same retention time in the shown HPLC traces.

We thank the reviewer for pointing out this. The overlapping is due to poor performance of the column. We have conducted the enzyme assays again. Please see the revised **Supplementary figure 4**.

Fig. 7: what is the peak at ~ 7.2 min?

We thank the reviewer for pointing out this, which may be contamination from the HPLC system. We have repeated the assays, please see the revised **Supplementary figure 9**.

Overall, this is a thoroughly conducted study leading to a data-rich manuscript, but this seems to be thrown together with data from various experiments, lacking depth in the key topic (VibO studies). As is, the manuscript might lack the originality and depth required for a top journal as Nature Communications. It might be better to finish the VibO studies and publish these alone, while the reconstitution pathway and other studies could be published elsewhere.

Nature evolves VibO in the *B. vibrans* mushroom to form the oxepinone as a stable metabolite and on-pathway precursor for the vibralactone pathway. Vibralactone has a unique chemical structure and potent biological activity. Identification of the oxepinone-forming VibO completes the biosynthesis of vibralactone. The vibralactone biosynthetic genes are unclustered and genome mining approaches are very limited. In addition, only a few have been done on the biosynthetic mechanisms for natural products isolated from mushrooms in general.

Specific responses to the criticisms from the Reviewer #3:

387155 Pan, Zeng et al

Recommendation: suitable for publication in nature communications with minor edits.

The authors present the complete biosynthesis of the fungal metabolite vibralactone. They isolate, express, and characterize key enzymes for prenylation and oxepinone formation, and reconstruct the pathway heterologously.

This study combines a tremendous amount of work that leads to the production of vibralactone. This biologically active beta lactone is produced by *Boreostereum vibrans*, a basidiomycete fungus, for which biosynthetic gene clusters are unlikely clustered and genome mining approaches are very limited.

Here, the team succeeded in purifying a Baeyer-Villiger oxygenase VibO, a key enzyme in the vibralactone biosynthesis which forms an oxepine intermediate, via enzyme- activity guided fractionation. They carefully biochemically characterize the unique enzyme VibO, obtain a crystal structure, create site mutants, and perform isotope feeding studies. Additionally, the prenyltransferases VibP1 and VibP2 as well as the reductase BvCAR are characterized and expressed in *e.coli*.

Finally, a combination of the five key biosynthetic enzymes in a one-pot reaction as well as biosynthetic pathway engineering in vivo in *e.coli* are successfully conducted to yield vibralactone. What a comprehensive study - Congratulations to the authors!

While the manuscript is generally clear and experiments well thought through, there are a few minor comments and questions that could improve the manuscript:

- Baeyer-Villiger oxygenase VibO represents a new enzyme which should be put into context of other fungal Baeyer-Villiger oxygenases like the aflatoxin oxygenase or others with regards to their crystal structure, active site amino acids, substrate specificity.

We thank the reviewer for the above constructive comment. Actually, until now, there is only one crystal structure of the fungal Baeyer-Villiger oxygenase BVMO_{AFL838} in its native form has been solved (PDB ID: 5J7X), and the crystal structure of aflatoxin oxygenase (AflN) has not been determined yet. Following the reviewer's nice suggestion, we have carefully compared the crystal structure, active site amino acids, and substrate specificity of VibO with the fungal Baeyer-Villiger oxygenase BVMO_{AFL838} as well as the bacterial Baeyer-Villiger oxygenase MtmOIV (see **Fig. IV** below, and **Supplementary Fig. 25b-c, Supplementary Fig. 39d-e** in the revised manuscript). Our structural comparison analyses showed that the overall structure and the related active site residues of VibO are significantly different from that of BVMO_{AFL838}, which lacks the C-terminal Rossmann fold sub-domain (see **Fig. IVa-b** below, and **Supplementary Fig. 25c, Supplementary Fig. 39d** in the revised manuscript). Although VibO and MtmOIV have similar sub-domain organizations and overall structural architectures (see **Fig. IVc** below, and **Supplementary Fig. 25b** in the revised manuscript), their active site residues and substrate-binding specificities are quite different (see **Fig. IVd** below, and **Supplementary Fig. 39e** in the revised manuscript). Therefore, VibO represents a new enzyme that is distinct from any currently known Baeyer-Villiger oxygenases. In addition, we have performed the phylogenetic analysis and sequence alignment (**Supplementary figures 21-23**).

Fig. IV: Structural comparison analyses of VibO with Baeyer-Villiger oxygenase BVMO_{AFL838} and MtmOIV. (a and c) The combined ribbon and stick representation showing the overall structure comparison of the monomeric VibO and the fungal Baeyer-Villiger oxygenase BVMO_{AFL838} (pink, PDB ID: 5J7X) (a), or the bacterial Baeyer-Villiger oxygenase MtmOIV (forest green, PDB ID: 4K5S) (c). (b and d) The ribbon-stick-ball representation showing the detailed structural comparison of the key active site residues of VibO with that of BVMO_{AFL838} (PDB ID: 5J7X) (b), or MtmOIV (PDB ID: 4K5S) (d). These structural analyses demonstrate that the active site residues and substrate-binding specificities of VibO are quite different from currently known fungal and bacterial Baeyer-Villiger oxygenases.

- The isotope feeding to verify the activity of VibO, figure 2D, should include MS/MS data to further verify the isotope location within the metabolite fragments. It is not quite clear if the isotope analysis is based on MS only or if the labelled compounds were isolated and characterized? ¹⁸O and D labels would be nicely visible in ¹³C or ¹H spectra. The caption for figure 2 should list supplemental figures 15 and 16 for 2D.

We thank the reviewer for this constructive comment. Following the reviewer's suggestion, we have done the scale-up labeling experiments and purified the ¹⁸O- and D-labelled **3** for ¹³C- and ¹H-NMR analyses, respectively. Please see the revised **Supplementary figures 6, 7, 15 and 16**. The caption for figure 2 has been revised to list Supplementary figures 6, 7, 17 and 18 for 2D.

- Feeding experiment Figure 6: 1mM **4a** and **5a** were added to the fungal cultures, what was the yield? Production analysis is based on HRMS only, quantification of metabolites should be added or explained why EIC was used only.

Indeed, quantification of metabolites may give more convincing analysis of production. Our previous work on feeding **6a** (4.92 g in 15 L of *B. vibrans* culture broth) led to the isolation of **3a** (34 mg) and **1a** (18 mg), and feeding [U-¹³C]-**2** (100 mg in 0.5 L of *B. vibrans* culture broth) to the isolation of **3** (12 mg, ~30% labeled)

and **1** (51 mg, ~43% labeled), which demonstrated the conversion of **2** to **6** for the vibrallactone (**1**) pathway. So in this study to determine the intermediates between **2** and **6**, we use HRMS with assistance of authentic standards (**6a**, **3a**, and **1a**) for analyzing **4a** and **5a** feedings. The intermediacy of **4** and **5** was corroborated by identification and characterization of the corresponding enzymes including VibP, BvCAR, and BvAR. We have provided the explain in the revised manuscript.

- The optical rotation was recorded at different concentrations for isolated **3** and enzyme produced **3**. This should have been done at the same concentration.

Yes, the optical rotation has been recorded again at the same concentration. Please see the revised **Supplementary figure 8**.

- The estimated total yields of **3** from in vitro and in vivo production should be listed to give context to the fungal, WT production.

Data has been presented in the revised **Supplementary figures 6 and 8**. The in vitro reaction (17 μ M VibO, 0.5 mM **6**, 0.1 mM NADPH) in 0.1 mL for 2 h was estimated to give ~2 μ g **3**. The whole-cell transformation (in vivo, *E. coli* cell culture 2.5 L) for 24 h gave ~12 mg **3** from feeding 90 mg of **6**. From *B. vibrans* mycelial culture broth of 20 L for 21 days at 25 °C, 46 mg **3** and 1.8 g **1** were obtained (reference 6 in the main text).

Minor comments:

- Supplement figure 1 states ‘dozens of milligrams’ please give a number. The chromatographic separation is quite unusual: Chromatographic separation of **6a**, **3a**, and **3a'** (an isomer of **3a**) was conducted with elution of 25% B over 4.8 min and 100% B over the next 3.2 min where A was H₂O and B was methanol. The mass spectrometer was run in positive ionization mode. Can the authors comment on why not a longer HPLC run was used or a flatter gradient was tried? In Supplement figure 4, the retention time of **6** and **3** is almost overlapping, while for other chromatograms (Supplemental figure 5), the separation is ok, although the same experimental settings are listed. In many figures, and for certain compounds, negative ionization mode was used.

We thank the reviewer for the constructive comments. Feeding **6a** (4.92 g in 15 L of *B. vibrans* culture) for isolation of **3a** (34 mg) and **1a** (18 mg) has been presented in the revised Supplementary figure 1. The overlapping in Supplementary figure 4 is due to poor performance of the column. We have repeated the assays with a new column, please see the revised **Supplementary figure 4**. The mass spectrometer can be run in either positive or negative ionization mode but cannot in both modes at the same time. Compounds **1**, **3**, and **3'** were hardly seen in negative mode; compounds **4**, **5**, **6**, **7**, and the internal standard are obvious in both modes but more in negative mode. To recognize **6** from **6'**, positive ionization mode had to be used because **6'** was barely detected in negative mode. A flatter gradient was tried before, but the separation is similar. This time, to detect the putative hydroxy-oxepinones and repeat some of assays, a longer HPLC was run for a better separation, please see the revised **Supplementary figures 9, 12, and 14**.

- Supplemental figure 7: dried for ‘two overnights’ please give precise time information.

The precise time is 36 h. We have repeated the assays and analyzed with a longer HPLC run, please see the revised **Supplemental figure 9**.

- Supplemental figure 8: caption states that 70 mg of **6** was converted, while the main text states 90mg?

Since isolation of the side product **7** was unsuccessful due to its very low production in the in vivo *E. coli* whole-cell transformation (**Supplemental figure 14**), we tried the in vitro reaction instead. The crude enzymes were prepared from the cleared lysates of the induced *E. coli* cells (15 L) and incubated with 70 mg **6** for 4 h at 28 °C. Subsequent extraction and purification yielded ~0.5 mg of the product for ¹H-NMR analysis (**Supplemental figure 10**).

• Supplement figure 16 does not show lines for some of the base ions.

We thank the reviewer for pointing out this, which has been revised in **Supplementary figure 18**.

• Supplement figure 19: I am not an expert in sedimentation coefficients, but a standard protein with known MW and resulting S value might be needed for context?

We thank the reviewer for pointing out this for us. Following the reviewer's suggestion, we have added a standard BSA protein with known molecular weight and resulting S value in the revised manuscript (see **Fig. V** below, and **Supplementary Fig. 24a** in the revised manuscript).

Fig. V: VibO forms a stable dimer in solution. The sedimentation velocity data of VibO and BSA proteins showing that VibO forms a stable dimer in solution. In this drawing, BSA is included as a standard control sample, and “MW” stands for molecular weight.

References

1. Ridder L, Mulholland AJ, Rietjens IMCM, & Vervoort J (2000) A quantum mechanical/molecular mechanical study of the hydroxylation of phenol and halogenated derivatives by phenol hydroxylase. *J Am Chem Soc* 122(36):8728-8738.
2. Enroth C, Neujahr H, Schneider G, & Lindqvist Y (1998) The crystal structure of phenol hydroxylase in complex with FAD and phenol provides evidence for a concerted conformational change in the enzyme and its cofactor during catalysis. *Structure* 6(5):605-617.

REVIEWERS' COMMENTS

Reviewer #1 (Remarks to the Author):

This paper is now much improved, following positive response to the collective remarks received.

Ultimately, the paper's focus is the pathway, with the enzyme of interest studied to some detail. The full uncovering of the mechanism is not within scope of this paper, and the authors have provide further initial data to support their hypothesis.

however, much of this remains circumstantial evidence, so i suggest the abstract sentence "Computational modeling, mutagenesis, and ultraviolet-visible spectra-based analyses uncover some key 36 residues that provide the active site geometry for VibO to conduct its function, and mechanistically support 37 that VibO uses the flavin-C4a-OO(H) intermediate for catalysis."

is replaced with "Computational modelling and solution studies provide insight into the likely VibO active site geometry, and suggest possible involvement of a flavin-C4a-OO(H) intermediate."

Reviewer #2 (Remarks to the Author):

The authors responded constructively to all my requests s well as to those of the other reviewers. This manuscript is now acceptable.

Reviewer #3 (Remarks to the Author):

The resubmitted manuscript is greatly improved and addressed all my points of critique. Noteworthy several new experiments were conducted to support their findings.

This is a very comprehensive study and I congratulate the authors on their work.

We thank the anonymous reviewers for their valuable comments leading to improvement of the manuscript.

Point-by-point responses to the reviewers' comments:

(Reviewers' comments are in **blue**, and our responses are in **black**)

Specific responses to the comments from the Reviewer #1:

This paper is now much improved, following positive response to the collective remarks received.

Ultimately, the paper's focus is the pathway, with the enzyme of interest studied to some detail. The full uncovering of the mechanism is not within scope of this paper, and the authors have provide further initial data to support their hypothesis.

however, much of this remains circumstantial evidence, so i suggest the abstract sentence "Computational modeling, mutagenesis, and ultraviolet–visible spectra-based analyses uncover some key residues that provide the active site geometry for VibO to conduct its function, and mechanistically support that VibO uses the flavin-C4a-OO(H) intermediate for catalysis." is replaced with "Computational modelling and solution studies provide insight into the likely VibO active site geometry, and suggest possible involvement of a flavin-C4a-OO(H) intermediate."

Thank you very much for your kind consideration on this matter. Following reviewer's comments, we have revised the **abstract**.

Specific responses to the comments from the Reviewer #2:

The authors responded constructively to all my requests as well as to those of the other reviewers. This manuscript is now acceptable.

Thank you very much.

Specific responses to the comments from the Reviewer #3:

The resubmitted manuscript is greatly improved and addressed all my points of critique. Noteworthy several new experiments were conducted to support their findings.

This is a very comprehensive study and I congratulate the authors on their work.

Many thanks.